# Omnipresent Yet Overlooked: Heat Kernels in Combinatorial Bayesian Optimization

**Colin Doumont**
ETH Zürich
University of Cambridge
Tübingen AI Center

**Victor Picheny**
Secondmind

**Viacheslav Borovitskiy**[‡]
ETH Zürich
University of Edinburgh

**Henry Moss**[‡]
University of Cambridge
Lancaster University

## Abstract

Bayesian Optimization (BO) has the potential to solve various combinatorial tasks, ranging from materials science to neural architecture search. However, BO requires specialized kernels to effectively model combinatorial domains. Recent efforts have introduced several combinatorial kernels, but the relationships among them are not well understood. To bridge this gap, we develop a unifying framework based on heat kernels, which we derive in a systematic way and express as simple closed-form expressions. Using this framework, we prove that many successful combinatorial kernels are either related or equivalent to heat kernels, and validate this theoretical claim in our experiments. Moreover, our analysis confirms and extends the results presented in `Bounce`: certain algorithms' performance decreases substantially when the unknown optima of the function do not have a certain structure. In contrast, heat kernels are not sensitive to the location of the optima. Lastly, we show that a fast and simple pipeline, relying on heat kernels, is able to achieve state-of-the-art results, matching or even outperforming certain slow or complex algorithms.

## 1 Introduction

Many real-world challenges can be framed as optimization problems, where the objective functions are expensive-to-evaluate and black-box. For this type of problems, Bayesian Optimization (BO) has emerged as a preferred approach, successfully solving problems from tuning machine-learning algorithms (Snoek et al., 2012) to designing detectors for particle physics (Cisbani et al., 2020).

Although BO has mostly been concerned with continuous domains, in recent years, multiple methods have been developed to extend BO to combinatorial domains, such as strings in genetic design (Moss et al., 2020) or graphs in neural architecture search (Ru et al., 2020b). These new methods often consist of a pipeline involving multiple components, making it hard to know how each individual component affects the overall empirical performance. For example, the pipelines from `CASMOPOLITAN` (Wan et al., 2021) and `COMBO` (Oh et al., 2019) are often chosen as baselines for other methods, but their components are never evaluated individually.

In a first attempt to bridge this gap, Dreczkowski et al. (2024) introduced a modular benchmark, dividing combinatorial BO pipelines into three key components: (i) combinatorial kernel, (ii) acquisition-function optimizer, and (iii) presence of trust region. Using this distinction, the authors are able

---

[‡]Equal supervision.

39th Conference on Neural Information Processing Systems (NeurIPS 2025).

to assess the performance of the individual components, rather than the full pipeline. However, it remains unclear how different combinatorial kernels (i.e. the first component) relate to one another and what properties are responsible for their success.

In this paper, we aim to provide a clearer picture of combinatorial kernels and unify them into a common framework based on heat kernels. First, we present the necessary background information on combinatorial BO and the associated combinatorial kernels (Section 2). To relate these kernels, we then present our unifying framework, based on heat kernels, as well as our generalizations and extensions thereof (Sections 3). Finally, we analyze and validate our theoretical framework using empirical experiments (Section 4), and conclude by discussing our contributions (Section 5).

## 2 Background and related work

In the next sections, we give a short introduction to (continuous) Bayesian optimization (Section 2.1) and combinatorial Bayesian optimization (Section 2.2), as well as provide an overview of existing combinatorial kernels (Section 2.3).

### 2.1 Bayesian optimization

Bayesian optimization (Garnett, 2023) aims to find the global optimum $\mathbf{x}^* \in \mathcal{X}$ of an expensive black-box function $f : \mathcal{X} \to \mathbb{R}$. BO constructs a probabilistic model of $f$, often using a Gaussian Process (GP), which estimates $f(\mathbf{x})$ at previously unseen locations $\mathbf{x} \in \mathcal{X}$, and quantifies uncertainty of such estimates. The probabilistic model is used to define an acquisition function $\alpha(\mathbf{x})$, which balances exploration and exploitation. The next point to evaluate is selected by maximizing the acquisition function:

$$\mathbf{x}_{t+1} = \arg\max_{\mathbf{x} \in \mathcal{X}} \alpha(\mathbf{x}).$$

By iteratively updating the model with new observations and selecting points that maximize $\alpha$, BO aims to efficiently converge to the global optimum, while minimizing the number of costly evaluations of $f$.

The GP is specified by a mean function and a covariance function or kernel $k(\mathbf{x}, \mathbf{x}') : \mathcal{X} \times \mathcal{X} \to \mathbb{R}$ (Rasmussen and Williams, 2006). When $\mathcal{X} \subseteq \mathbb{R}^n$, a common choice is the RBF kernel:

$$k_{\ell_i}(\mathbf{x}, \mathbf{x}') = \exp\left(-\frac{1}{2}\sum_{i=1}^{n} \frac{(x_i - x_i')^2}{\ell_i}\right),$$

where $\ell_i > 0$ are the lengthscales. By assigning one lengthscale per dimension rather than a single global lengthscale, the model can automatically assess the importance of each input dimension. This feature is also known as Automatic Relevance Determination (ARD).

### 2.2 Combinatorial Bayesian optimization

In this paper, instead of considering Euclidean input spaces, we assume $\mathcal{X}$ is a product $\mathcal{X} = \times_{i=1}^{n} \mathcal{X}_i$ of finite sets $\mathcal{X}_i$. By defining $\mathcal{X}$ in this way, we encompass many domains of interest for real-world applications, such as categorical variables, strings, or even graphs. However, we emphasize that the above definition of $\mathcal{X}$ does not encompass ordinal variables, since we define $\mathcal{X}_i$ as sets (unordered) and not sequences (ordered). As a result, all theorems in this paper hold for categorical variables, but not necessarily for ordinal variables. Nevertheless, in Appendix B, we argue that *truly* ordinal variables are a a rare exception in combinatorial Bayesian optimization, and the results in this paper thus apply to nearly all previous works in the field.

Performing Bayesian optimization on this combinatorial input space $\mathcal{X}$ consists of two major challenges: *surrogate modeling* and *acquisition-function optimization*. Often, the two challenges are treated separately, with different combinatorial BO methods having either the same surrogate model or the same acquisition-function optimizer (Dreczkowski et al., 2024). In this work, we only focus on surrogate modeling and use pre-existing methods for acquisition-function optimization.

There are three primary ways to define surrogate models on combinatorial spaces:

1. **Use inherently discrete models**, such as random forests in `SMAC` (Hutter et al., 2011), the Tree-structured Parzen Estimator in `TPE` (Bergstra et al., 2011) or a sparse Bayesian linear model in `BOCS` (Baptista and Poloczek, 2018).

2. **Map input points to a Euclidean space**, after which a continuous model (often a GP) can be used. This is usually done for tasks where there already exists a large library of relevant structures, enabling deep generative models to create a continuous latent space (Griffiths and Hernández-Lobato, 2020; Moss et al., 2025). A common example of such a library is the ZINC database (Irwin and Shoichet, 2005), which is a curated collection of commercially available chemical compounds.

3. **Use combinatorial kernels**, which we define as kernels $k(\mathbf{x}, \mathbf{x}') : \mathcal{X} \times \mathcal{X} \to \mathbb{R}$ that directly take elements in a combinatorial space as input. Through the development of custom-made combinatorial kernels for specific tasks, GPs have successfully achieved state-of-the-art performance on Bayesian optimization problems across different combinatorial domains (Ru et al., 2020b; Moss et al., 2020; Tripp and Hernández-Lobato, 2024).

For tasks where little or no prior information is available, GPs with combinatorial kernels have consistently outperformed other models (Papenmeier et al., 2023; Dreczkowski et al., 2024), and are therefore the focus of this paper. We present a short overview of combinatorial kernels in the following section.

## 2.3 Kernels on combinatorial spaces

The Hamming distance is a recurring element in the literature on combinatorial kernels, because it represents a natural way of defining the similarity between two points $\mathbf{x}$ and $\mathbf{x}'$ in the combinatorial space $\mathcal{X}$. Essentially, the Hamming distance measures the minimum number of substitutions required to change one vector $\mathbf{x}$ into the other vector $\mathbf{x}'$. More formally, we define the Hamming distance as $h(\mathbf{x}, \mathbf{x}') = n - \sum_{i=1}^{n} \delta(x_i, x_i')$, where $\mathbf{x}, \mathbf{x}' \in \mathcal{X}$ are vectors of size $n$ and $\delta(\cdot, \cdot)$ is the Kronecker delta function. Therefore, the Hamming distance is in fact nothing more than one-hot encoding the data followed by the squared Euclidean distance (divided by two). That is,

$$h(\mathbf{x}, \mathbf{x}') = \|\mathbf{z} - \mathbf{z}'\|_2^2 / 2, \tag{1}$$

where $\mathbf{z}, \mathbf{z}' \in \{0, 1\}^s$ are the one-hot encoded versions of $\mathbf{x}, \mathbf{x}' \in \mathcal{X}$, with $s = \sum_{i=1}^{n} |\mathcal{X}_i|$. Equation 1 is a first step in providing a more unified view on combinatorial kernels, and is implicitly used in the following paragraph to present existing combinatorial kernels in terms of only the Hamming distance, rather than a mixture of Hamming and Euclidean distances (after one-hot encoding).

**Hamming-based kernels** A large proportion of existing combinatorial kernels rely on the Hamming distance in one form or another. For example, `CASMOPOLITAN` and `Bounce` use the well-known RBF and Matérn kernels, respectively, but replace the squared Euclidean distance by the Hamming distance (Wan et al., 2021; Papenmeier et al., 2023). Rather than using the Hamming distance directly, `COMBO` proposes to model $\mathcal{X}$ as a Hamming graph $\mathcal{G} = (\mathcal{X}, \mathcal{E})$ (Imrich et al., 2000), where the nodes $\mathbf{x}, \mathbf{x}' \in \mathcal{X}$ are connected by an edge only when they have a Hamming distance of one (Oh et al., 2019). Once $\mathcal{X}$ has been transformed into $\mathcal{G}$, a graph kernel (Kondor and Lafferty, 2002) is applied on $\mathcal{G}$. Although this graph-based approach appears different to the more direct approach taken in `CASMOPOLITAN` and `Bounce`, we show in Section 3 that both approaches are closely related. Similarly to `COMBO`, Deshwal et al. (2023) also propose to first transform the input space $\mathcal{X}$ before applying a kernel on this transformed space. More specifically, the authors introduce the idea of a Hamming Embedding via Dictionaries (`HED`), which embeds categorical inputs into an ordinal feature space, on which standard continuous kernels can be applied. They name the method `BODi`.

**Kernels for specific domains** Some state-of-the-art kernels have been developed with specific spaces of $\mathcal{X}$ in mind. For example, for string spaces, Moss et al. (2020) introduce `BOSS`, which uses an efficient dynamic programming algorithm to compute the Sub-sequence String Kernel (`SSK`) from Lodhi et al. (2002). Similarly, for graphs, Ru et al. (2020b) develop `NASBOWL`, which uses the Weisfeiler–Lehman (`WL`) kernel from Shervashidze et al. (2011) to perform neural architecture search.

# 3   Unifying framework

In this section, we show that most of the general-purpose kernels from the combinatorial BO literature are in fact closely related or even identical. Specifically, we provide a unifying framework based on heat kernels (Section 3.1), showing that most combinatorial kernels are either heat kernels (Sections 3.2 and 3.3) or part of a certain generalized class containing heat kernels (Section 3.4). Moreover, we extend our heat-kernel framework to group invariances and additive structures (Section 3.5).

## 3.1   Heat kernels for combinatorial Bayesian optimization

Heat kernels, also called diffusion kernels, are a natural family of kernels *on* graphs (but not *between* graphs). More specifically, these graph kernels are based on the heat equation and can be regarded as the discrete counterparts of the familiar RBF kernel over Euclidean spaces. We refer readers to Kondor and Lafferty (2002) for a more detailed explanation.

Before defining heat kernels more formally, we first clarify the necessary terminology. For the remainder of this paper, when we consider an arbitrary graph $\mathcal{G} = (\mathcal{V}, \mathcal{E})$, we assume this graph to be unweighted, undirected and without loops. Additionally, we define the graph Laplacian of $\mathcal{G}$ as $\mathbf{\Delta} = \mathbf{D} - \mathbf{A}$, where $\mathbf{A}$ is the adjacency matrix and $\mathbf{D}$ is the diagonal degree matrix, namely $\mathbf{D}_{ii} = \sum_{j=1}^{|\mathcal{V}|} \mathbf{A}_{ij}$. Using the graph Laplacian, we can define heat kernels as follows.

**Definition 1.** Let $(\lambda_j, f_j)$ be the eigenpairs of the Laplacian of a graph $\mathcal{G} = (\mathcal{V}, \mathcal{E})$, then the *heat kernel* is given by

$$k_\beta(x, x') = \sum_{j=1}^{|\mathcal{V}|} e^{-\beta \lambda_j} f_j(x) f_j(x'), \tag{2}$$

where $x, x' \in \mathcal{V}$ and $\beta > 0$ is a hyperparameter.

**Transforming $\mathcal{X}$ into $\mathcal{G}$**   Since we defined the input space $\mathcal{X}$ as being the product of finite sets $\mathcal{X}_i$ (see Section 2.2), we can naturally view $\mathcal{X}$ as being a product graph $\mathcal{G}$, where each subgraph $\mathcal{G}_i = (\mathcal{X}_i, \mathcal{E}_i)$ contains the elements of $\mathcal{X}_i$ as nodes. To specify the set of edges $\mathcal{E}_i$ between these nodes, different options may seem natural depending on the setting at hand. Possible examples include cycle graph or simple path structures, but, for our setting, perhaps the most straightforward are complete graphs. By connecting all elements in $\mathcal{X}_i$, we explicitly incorporate the fact that all elements are "equally distant", which seems reasonable when we have no prior information about $\mathcal{X}_i$. We call this transformed input space the Hamming graph $\mathcal{G}$ and define it as follows.

**Definition 2.** Assume $\mathcal{X}$ is an $n$-dimensional space that decomposes as $\times_{i=1}^{n} \mathcal{X}_i$, where each $\mathcal{X}_i$ is a finite set. Then, we can transform $\mathcal{X}$ into a corresponding *Hamming graph* $\mathcal{G} = (\mathcal{X}, \mathcal{E})$ by

$$\mathcal{G} = \mathop{\square}_{i=1}^{n} \mathcal{G}_i,$$

where $\square$ is the graph Cartesian product and $\mathcal{G}_i = (\mathcal{X}_i, \mathcal{E}_i)$ are complete unweighted graphs (Imrich et al., 2000).

Kondor and Lafferty (2002) show that, when applied to the Hamming graph of Definition 2, the heat kernel from Equation 2 simplifies to

$$k_\beta(x, x') \propto \prod_{i=1}^{n} \left[ \frac{1 - e^{-\beta g_i}}{1 + (g_i - 1)e^{-\beta g_i}} \right]^{1 - \delta(x_i, x_i')}, \tag{3}$$

where $g_i = |\mathcal{X}_i|$ and $\beta > 0$ is the same hyperparameter as in Equation 2. Additionally, when $\beta$ is replaced with $\beta_i$, we obtain the ARD version. Among others, the above simplification is due to the graphs $\mathcal{G}_i$ being complete, which means the eigenpairs can be computed analytically, and there is no need for expensive numerical eigendecomposition.

In Appendix C, we argue that heat kernels provide a natural measure of covariances on combinatorial spaces, arising from two natural explicit assumptions. Furthermore, in Appendix D.2, we demonstrate that relaxing one of these assumptions allows us to easily extend heat kernels to additive structures, connecting them to well-known combinatorial kernels.

## 3.2 CASMOPOLITAN and COMBO are heat kernels

CASMOPOLITAN (Wan et al., 2021) and COMBO (Oh et al., 2019) are two established methods for combinatorial BO, and are included in all recent benchmarks (Deshwal et al., 2023; Papenmeier et al., 2023; Dreczkowski et al., 2024). For surrogate modeling, both methods use a GP and introduce the following combinatorial kernels.

**Definition 3.** Let $\delta(\cdot, \cdot)$ be the Kronecker delta function, then the *CASMOPOLITAN kernel* is given by

$$k_{\ell_i}(\mathbf{x}, \mathbf{x}') = \exp\left(\frac{1}{n} \sum_{i=1}^{n} \ell_i \, \delta\left(x_i, x_i'\right)\right), \tag{4}$$

where $\mathbf{x}, \mathbf{x}' \in \mathcal{X}$ and $\ell_i > 0$ are hyperparameters (the lengthscales).

As hinted at in Section 2.3, CASMOPOLITAN can be seen as the analog of the RBF kernel, where the squared Euclidean distance is replaced by the ARD version of the Hamming distance. In fact, by virtue of Equation 1, the non-ARD version of CASMOPOLITAN is equivalent to one-hot encoding $\mathbf{x}, \mathbf{x}' \in \mathcal{X}$ and using the RBF kernel.

**Definition 4.** Let $\mathcal{G} = (\mathcal{X}, \mathcal{E})$ be a Hamming graph as in Definition 2 and $\left(\lambda_j^i, f_j^i\right)$ be the eigenpairs of the Laplacian of the graph $\mathcal{G}_i = (\mathcal{X}_i, \mathcal{E}_i)$, then the *COMBO kernel* is given by

$$k_{\beta_i}(\mathbf{x}, \mathbf{x}') = \prod_{i=1}^{n} \sum_{j=1}^{|\mathcal{X}_i|} e^{-\beta_i \lambda_j^i} f_j^i\left(x_i\right) f_j^i\left(x_i'\right), \tag{5}$$

where $\mathbf{x}, \mathbf{x}' \in \mathcal{X}$ and $\beta_i > 0$ are hyperparameters.

Although at first sight the kernels from CASMOPOLITAN and COMBO might seem like two separate methods, they are in fact closely related and can both be seen as heat kernels on a Hamming graph. We formalize this statement in the following theorem.

**Theorem 5.** *The kernels from CASMOPOLITAN and COMBO are equivalent to heat kernels on a Hamming graph. That is, Equations 4 and 5 are proportional to (the ARD version of) Equation 3.*

*Proof.* See Appendix A.1. □

### 3.3 New insights into combinatorial kernels

Theorem 5 results in multiple new insights, both theoretical and practical.

#### 3.3.1 CASMOPOLITAN and COMBO rely on the same kernel

This result follows directly from Theorem 5, since Equation 4 and 5 are now also equivalent. To the best of our knowledge, this equivalence between the kernels of CASMOPOLITAN and COMBO has never been established before, with most recent papers using the two methods on the same benchmarks (Deshwal et al., 2023; Papenmeier et al., 2023). However, because both kernels are often benchmarked with markedly different hyperparameter priors and acquisition-function optimizers, sometimes even with implementation errors (Papenmeier et al., 2023), this theoretical equivalence has never been visible empirically (Dreczkowski et al., 2024). In Section 4, we control for these different factors, and demonstrate that this theoretical equivalence also translates to an empirical equivalence. That is, the kernels from CASMOPOLITAN and COMBO, as well as the heat kernel (on a Hamming graph) from Equation 3, obtain the same performance in our experiments.

#### 3.3.2 Heat kernels (on Hamming graphs) are RBF kernels after one-hot encoding

Combining Theorem 5 and Equation 1, we have: heat kernels (on Hamming graphs) are equivalent to CASMOPOLITAN's kernel, which is itself equivalent to the RBF kernel after one-hot encoding. This connection means that heat kernels (on Hamming graphs) can easily be implemented, requiring only one additional line to one-hot encode the data before using standard implementations of the RBF kernel. For the ARD version, however, this equivalence does not hold, and Equation 3 should be used instead. Borovitskiy et al. (2023) demonstrate a similar relationship for binary variables, but, to the best of our knowledge, the extension to categorical variables has never been shown before.

### 3.3.3 Faster implementation of `COMBO` (for categorical variables)

The existing implementations of `COMBO` (Oh et al., 2019; Dreczkowski et al., 2024) currently happen as follows: (i) perform numerical eigendecomposition to obtain the eigenpairs $\left(\lambda_j^i, f_j^i\right)$, and (ii) use these eigenpairs to compute Equation 5. Instead, Theorem 5 presents a faster solution: use Equation 4 (or the ARD version of Equation 3), which implicitly computes the analytic eigendecomposition, rather than relying on expensive numerical eigendecomposition. As a result, the current implementation of `COMBO`'s kernel can be improved from $\mathcal{O}\left(\sum_{i=1}^n |\mathcal{X}_i|^3\right)$ to $\mathcal{O}(n)$. For our experiments in Section 4, this improved implementation resulted in a more than $2\times$ improvement in speed (see Appendix E.6). Although this result is only true for categorical variables, we note that most recent benchmarks in combinatorial Bayesian optimization do not contain any ordinal variables (Deshwal et al., 2023; Papenmeier et al., 2023; Dreczkowski et al., 2024).

## 3.4 Generalizing heat kernels: Hamming- and graph-based approaches

This new connection between the Hamming-based approach from `CASMOPOLITAN` and the graph-based approach from `COMBO` leads us to wonder how closely related these two approaches are. In the next sections, we formally define both approaches and show that they are deeply connected.

### 3.4.1 Hamming-based approach

To generalize `CASMOPOLITAN` into a broader class of kernels, we identify its main idea: leveraging a well-known continuous (isotropic) kernel (i.e. the RBF kernel), but substituting the standard Euclidean distance with the square root of the Hamming distance. Based on this foundational idea, we define the following class of kernels.

**Definition 6.** We define *Hamming kernels* as being kernels who only depend on $\mathbf{x}$ and $\mathbf{x}'$ through the square root of the Hamming distance $h(\mathbf{x}, \mathbf{x}')$, namely

$$k(\mathbf{x}, \mathbf{x}') = \Bbbk \left( \sqrt{h(\mathbf{x}, \mathbf{x}')} \right),$$

for some $\Bbbk : \{0, \sqrt{1}, \sqrt{2}, \dots, \sqrt{n}\} \to \mathbb{R}$.

Examples of Hamming kernels include `CASMOPOLITAN` and `Bounce`, and can be obtained by selecting the following functions for $\Bbbk$:

$$\Bbbk_\ell(d) = \exp\left(-\frac{d^2}{\ell^2}\right)$$

for `CASMOPOLITAN`, or

$$\Bbbk_\ell(d) = \left(1 + \frac{\sqrt{5}d}{\ell} + \frac{5d^2}{3\ell^2}\right) \exp\left(-\frac{\sqrt{5}d}{\ell}\right)$$

for `Bounce`, with $d := \sqrt{h(\mathbf{x}, \mathbf{x}')}$ and $\ell > 0$ a hyperparameter. In fact, as we will show in the following proposition, by choosing $\Bbbk(d)$ to be any (continuous) isotropic kernel (with $d = \|\mathbf{x} - \mathbf{x}'\|_2$), we always obtain a valid Hamming kernel $k$.

**Proposition 7.** *Let $k : \mathbb{R}^n \times \mathbb{R}^n \to \mathbb{R}$ be symmetric positive semi-definite and*

$$k(\mathbf{z}, \mathbf{z}') = \Bbbk\left(\|\mathbf{z} - \mathbf{z}'\|_2\right), \quad \text{with } \mathbf{z}, \mathbf{z}' \in \mathbb{R}^n,$$

*for some $\Bbbk : \mathbb{R} \to \mathbb{R}$. Then, the kernel*

$$k^*(\mathbf{x}, \mathbf{x}') = \Bbbk\left(\sqrt{h(\mathbf{x}, \mathbf{x}')}\right), \quad \text{with } \mathbf{x}, \mathbf{x}' \in \mathcal{X},$$

*is always symmetric positive semi-definite.*

*Proof.* See Appendix A.2. □

Using Proposition 7, we arrive at a general procedure for generating new combinatorial kernels: choose any (continuous) isotropic kernel and replace the Euclidean distance by the square root of the Hamming distance. For instance, an interesting new kernel can be constructed by defining $\Bbbk$ as the well-known rational quadratic kernel (Rasmussen and Williams, 2006).

### 3.4.2 Graph-based approach

To generalize `COMBO` into a broader class of kernels, we identify its main idea: transforming the input space $\mathcal{X}$ into a Hamming graph $\mathcal{G}$, and then applying a graph kernel. While numerous graph kernels exist, one prominent family is the class of $\Phi$-kernels (Smola and Kondor, 2003; Borovitskiy et al., 2023), which define the kernel in terms of a regularization function $\Phi(\lambda)$ of the graph's eigenvalues.

**Definition 8.** Let $(\lambda_j, f_j)$ be the eigenpairs of the Laplacian of a graph $\mathcal{G} = (\mathcal{V}, \mathcal{E})$, then $\Phi$-*kernels* are given by

$$k_\beta\left(x, x'\right) = \sum_{j=1}^{|\mathcal{V}|} \Phi_\beta(\lambda_j) \, f_j\left(x\right) f_j\left(x'\right), \tag{6}$$

where $x, x' \in \mathcal{V}$ and $\Phi_\beta : \mathbb{R} \to [0, \infty)$, with $\beta$ a (set of) hyperparameter(s).

By selecting $\Phi_\beta\left(\lambda\right) = \exp\left(-\beta\lambda\right)$ in Equation 6 and applying the simplification from Kondor and Lafferty (2002), we recover Equation 5 (without ARD), demonstrating that `COMBO` is indeed (the ARD version of) a $\Phi$-kernel. Additionally, this connection with $\Phi$-kernels is yet another way to come up with new kernels. By transforming $\mathcal{X}$ into $\mathcal{G}$ and selecting any function $\Phi : \mathbb{R} \to [0, \infty)$, we arrive at a valid combinatorial kernel $k$, which is again guaranteed to be symmetric and positive semi-definite. Rather than coming up with new definitions of $\Phi\left(\lambda\right)$, further work could first try well-known, state-of-the-art $\Phi$-kernels, such as the graph Matérn kernel (Borovitskiy et al., 2021) or the sum-of-inverse polynomial kernel (Wan et al., 2024).

### 3.4.3 Equivalence of Hamming- and graph-based approaches

Because of Theorem 5, we know there is at least one kernel from the class of Hamming kernels, namely `CASMOPOLITAN`, that corresponds to an equivalent kernel in the class of $\Phi$-kernels, namely `COMBO`. In fact, we show in Theorem 9 that, for finite sets $\mathcal{X}_i$ of equal size, any Hamming kernel must be a $\Phi$-kernel for some choice of $\Phi$, and vice versa. Note that Theorem 9 does not hold for finite sets of different sizes; see Appendix A for a counter example. Although this assumption might seem restrictive, we point out that it still holds for every single problem considered in recent benchmarks (Deshwal et al., 2023; Papenmeier et al., 2023; Dreczkowski et al., 2024), including this paper.

**Theorem 9.** *Let $\mathcal{G} = (\mathcal{X}, \mathcal{E})$ be a Hamming graph with $|\mathcal{X}_1| = \cdots = |\mathcal{X}_n|$. Then, the class of Hamming kernels on $\mathcal{X}$ coincides with the class of $\Phi$-kernels on $\mathcal{G}$.*

*Proof.* See Appendix A.3. $\qquad\square$

Theorem 9 shows that, for finite sets of equal size, Hamming- and graph-based approaches are fundamentally the same, and that using one amounts to using the other. This equivalence appears to be new, as most combinatorial BO papers present `CASMOPOLITAN` and `COMBO` as using different approaches (e.g. Deshwal et al., 2023), and not different perspectives of the same approach.

### 3.5 Extending heat kernels: incorporating invariances and additive structures

**Group invariances** In Appendix D.1, we review different methods to incorporate *any* group invariance within heat kernels. Moreover, we propose a simple trick that significantly improves performance on permutation-invariant functions, and matches the `SSK` with only a fraction of the wall-time and compute (Figure 8). Lastly, using group-invariant heat kernels, we match the `WL` graph kernel on a set of neural-architecture-search problems (Figure 10), showcasing the advantage of incorporating group invariances for real-world tasks.

**Additive structures** In Appendix D.2, we use our systematic derivation from Appendix C to establish a clear connection between heat kernels and additive-based kernels, such as `CoCaBO` and `RDUCB`, as well as propose a new "explainable" kernel. Although incorporating additive structure often decreases performance for standard benchmarks (Figures 1 and 4), we note a clear improvement in performance when applied to biological datasets (Figures 6 and 7), which we discuss in Appendix E.4.

Although the above extension allows us to connect heat kernels to an even wider group of well-known kernels—now also containing `CoCaBO` and `RDUCB`—we argue that `BODi`'s `HED` kernel is still fundamentally different from heat kernels (or variations thereof), and provide a proof in Appendix A.4.

# 4 Experiments

**Benchmarks and baselines**   Our analysis is performed on a wide range of challenging combinatorial optimization problems: `Pest Control`, `LABS` and `Cluster Expansion` are displayed below in Figures 1 and 2, and `Contamination Control` and `MaxSAT` have been moved to Appendix E.3 (due to space constraints). All five problems are taken from `Bounce` (Papenmeier et al., 2023), and are described in more detail in Appendix E.1. Additionally, in Appendices E.4 and E.5, we experiment on five (discretized) permutation-invariant Simon Fraser University (SFU) test functions (Surjanovic and Bingham, 2013), as well as two biological and two logic-synthesis tasks from the `MCBO` benchmark.

We compare against several competitive baselines, including: `COMBO` (Oh et al., 2019), `CoCaBO` (Ru et al., 2020a), `BOSS` (Moss et al., 2020), `CASMOPOLITAN` (Wan et al., 2021), `BODi` (Deshwal et al., 2023), `RDUCB` (Ziomek and Ammar, 2023) and `Bounce` (Papenmeier et al., 2023). All baselines were evaluated using their corresponding implementation in the `MCBO` benchmark, except for `Bounce`, which is not present in `MCBO` and was evaluated using the authors' original implementation.

**Experimental set-up**   In Figure 1, we keep a fixed pipeline and only change the combinatorial kernel, so that differences in other (confounding) factors of the pipeline do not affect results. Based on the ablation study from the `MCBO` paper, we choose the following pipeline: the genetic algorithm used in `BOSS` (Moss et al., 2020), together with the trust region from `CASMOPOLITAN` (Wan et al., 2021). In Figure 2, we combine this pipeline with a heat kernel, and compare it against other, well-known pipelines. In both figures, and as done in `Bounce` (Papenmeier et al., 2023), we display the originally published formulation, as well as a modification with the optimal point moved to a random location (marked by [reloc.]). All methods have been evaluated for at least 20 different random seeds, with more noisy datasets having up to 60 seeds (e.g. the biological tasks from Appendix E.4). Further details regarding our experimental set-up are described in Appendix E.2, and our implementation can be found at: https://github.com/colmont/heat-kernels-4-BO.git.

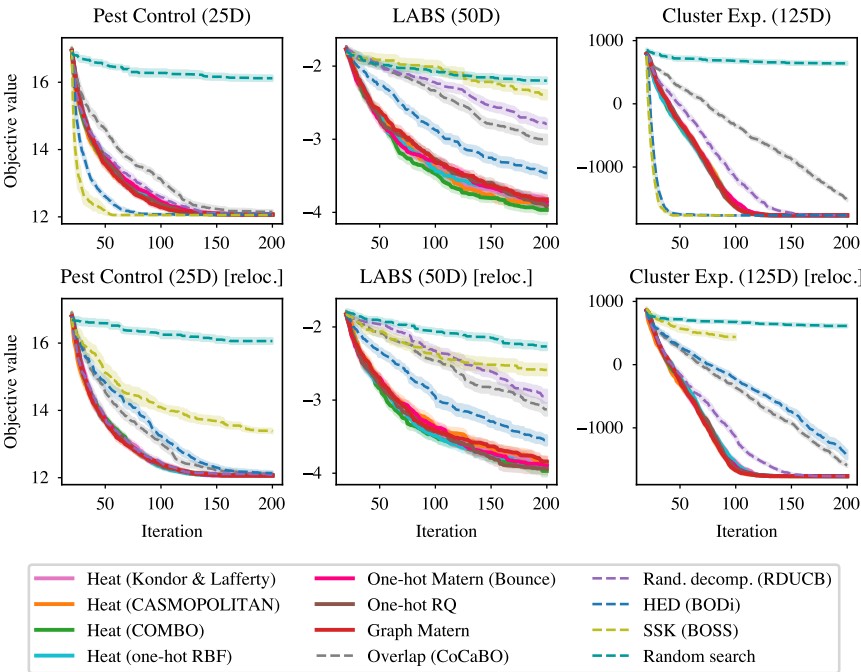

Figure 1: When keeping the pipeline fixed and varying only the kernel choice, our unifying framework becomes visible empirically: all heat (and Hamming) kernels achieve near-identical results.

## 4.1   Empirical evidence for the theoretical claims of the unifying framework

In Sections 3.1-3.3, we proved that the following four kernels are *theoretically equivalent*: the RBF kernel (after one-hot encoding), and the kernels found in Kondor & Lafferty (Equation 3),

`CASMOPOLITAN` (Equation 4) and `COMBO` (Equation 5). Additionally, in Section 3.4, we argued that these four kernels are closely related—through the concept of "Hamming kernels"—to `Bounce`'s Matérn kernel (after one-hot encoding), the graph Matérn kernel (Borovitskiy et al., 2021), and the Rational Quadratic (`RQ`) kernel after one-hot encoding (proposed in Section 3.4.2). Now, we show that these seven kernels are also *empirically equivalent*: on all problems from Figures 1, 4 and 6, the seven kernels (indicated by a solid line) achieve nearly indistinguishable performance.

## 4.2    Sensitivity of `BODi` and `BOSS` to the location of the optima

In Figures 1 and 4, `BODi`'s `HED` kernel and `BOSS`' SSK achieve significantly worse results when the optima are relocated. For `BODi` (`HED`), Papenmeier et al. (2023) provide an extended analysis, arguing that this behavior is due to the kernel's higher probability of sampling the last category, which is often the most "rewarding" category (at least in the original version of the above problems). For `BOSS` (SSK), we believe we are the first to report this behavior, and attribute it to a misuse of the kernel: the SSK was designed specifically for strings, and not arbitrary categorical inputs (e.g. as is done in `MCBO`). Specifically, we hypothesize that the SSK performs well because it favors points with repeated categorical values, which are more "rewarding" in the original version of the above problems (but not necessarily in the modified ones, hence the performance degradation). We provide a more detailed explanation in Appendix E.5. In contrast, heat (and Hamming) kernels, as well as the related `CoCaBO` and `RDUCB` kernels (see Appendix D.2), are unaffected when the optimum is relocated.

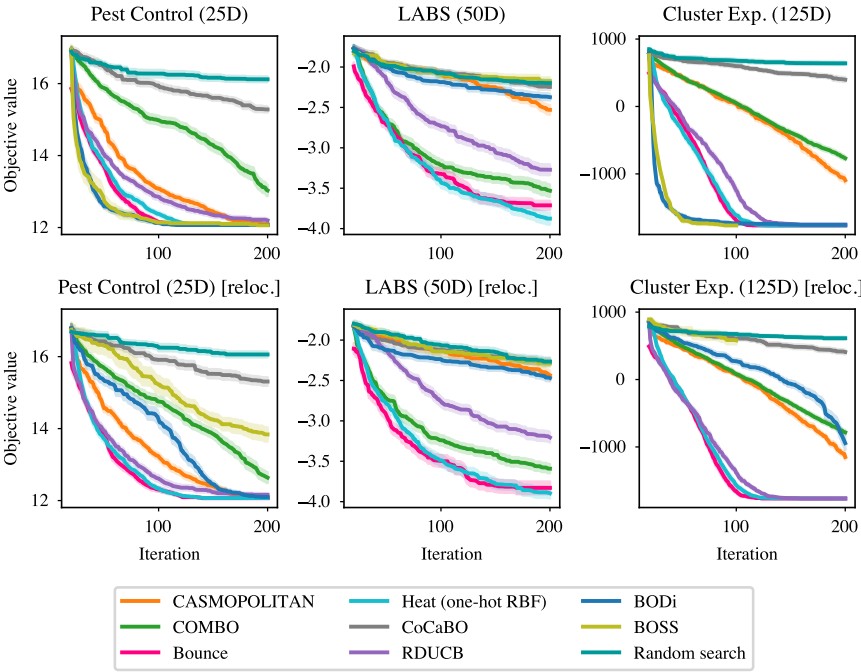

Figure 2: A fast and simple pipeline, relying on heat kernels, achieves state-of-the-art results (after relocation of the optima), matching or even outperforming more complex or slow baselines.

## 4.3    Fast and simple heat-kernel pipeline with state-of-the-art performance

In Figures 2 and 5, all baselines are matched or outperformed (after relocation of the optima) by our heat-kernel pipeline, which consists of three components: (i) the RBF kernel (after one-hot encoding), (ii) a genetic algorithm, and (iii) a trust region. We argue that this pipeline is *simple*, relying on established concepts from different fields, without introducing any new, convoluted ideas. In contrast, `Bounce`'s sophisticated nested-embeddings, or `RDUCB`'s tree-decomposition with message-passing, might make these methods less approachable for industry practitioners. Furthermore, this simple pipeline is *fast*, achieving the third lowest wall-clock time from all the evaluated baselines, closely following `CoCaBO` and `Bounce` (see Figure 3, with details in Appendix E.6).

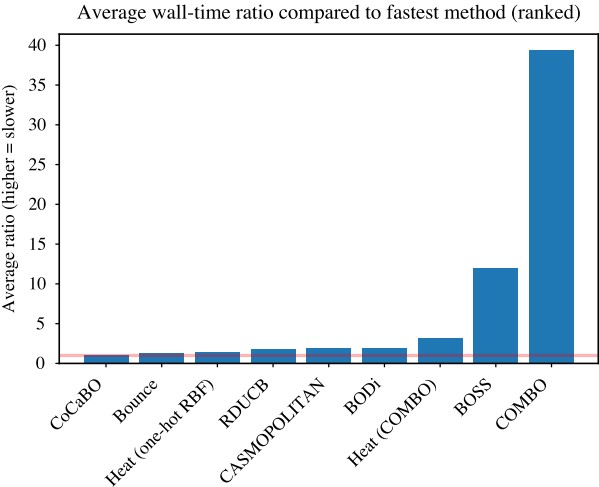

Figure 3: Our simple heat-kernel pipeline achieves the third-lowest wall-clock time, arriving closely after `CoCaBO` and `Bounce`.

## 5 Discussion and conclusion

In this paper, we have related different well-known combinatorial BO methods by presenting a unifying framework, based on heat kernels. This framework aims to provide a clearer picture of the existing combinatorial kernels, leading to new theoretical and practical insights.

Among others, we prove that `CASMOPOLITAN` and `COMBO` are equivalent to heat kernels (on Hamming graphs), and that both methods amount to using the RBF kernel (after one-hot encoding). By generalizing heat kernels, we establish new links with existing combinatorial kernels (e.g. `Bounce`) and obtain a principled framework for generating new combinatorial kernels, guaranteed to be symmetric positive semi-definite. Similarly, by extending our framework with additive structure and group-invariance, we can again link heat kernels to well-known combinatorial kernels (e.g. `CoCaBO` and `RDUCB`), and achieve competitive performance on a real-world permutation-invariant task (e.g. neural architecture search).

In our experiments, we validate these theoretical claims with empirical evidence, showing that all the above heat (and even Hamming) kernels achieve near-identical results on five different problems. Moreover, we are the first to observe the `SSK`'s degradation after relocation of the optima, extending the recent results found in `Bounce` (Papenmeier et al., 2023). Lastly, we propose a simple and fast heat-kernel pipeline with state-of-the-art results, hopefully making combinatorial BO more accessible.

**Limitations** While heat kernels show great performance on all of the evaluated problems, we emphasize that this is only true for settings with *little or no a priori information*. When there already exists a large library of relevant structures, or when the inputs are known to have certain properties (e.g. molecules), other methods might be preferred. Moreover, our pipeline only considers categorical variables, although the extension to mixed variables (i.e. categorical and continuous) is straightforward (see Ru et al., 2020a; Papenmeier et al., 2023, for example). Lastly, our experiments do not contain (truly) ordinal variables, but we note that this is standard in recent combinatorial BO papers (Deshwal et al., 2023; Papenmeier et al., 2023).

**Societal impact** Although our main contribution is theoretical (i.e. a unifying framework), it does lead to a number of practical improvements for combinatorial Bayesian optimization. For example, the simple pipeline proposed in Section 4.3 could facilitate the use of combinatorial BO by a wider public. Similarly, the faster implementation described in Section 3.3.3 leads to similar results with lower computational resources, increasing its accessibility. However, we believe these improvements carry minimal risks beyond the standard concerns about possible misuse of the underlying technology.

**Acknowledgments**

VB was supported by ELSA (European Lighthouse on Secure and Safe AI) funded by the European Union under grant agreement No. 101070617. HM was supported by Schmidt Sciences and Research England under the Expanding Excellence in England (E3) funding stream, which was awarded to MARS: Mathematics for AI in Real-world Systems in the School of Mathematical Sciences at Lancaster University.

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

# A Proofs

## A.1 Proof of Theorem 5

**Theorem 5.** *The kernels from* `CASMOPOLITAN` *and* `COMBO` *are equivalent to heat kernels on a Hamming graph. That is, Equations 4 and 5 are proportional to (the ARD version of) Equation 3.*

*Proof.* For `CASMOPOLITAN`, we can rewrite Equation 4 as

$$k\left(\mathbf{x}, \mathbf{x}'\right) = \exp\left(\frac{1}{n}\sum_{i=1}^{n} \ell_i\, \delta\left(x_i, x_i'\right)\right) = \prod_{i=1}^{n} \exp\left(\gamma_i\, \delta\left(x_i, x_i'\right)\right),$$

where $\gamma_i = \frac{\ell_i}{n}$ and $\ell_i > 0$ are hyperparameters. Now, if we define $\gamma_i$ as

$$\gamma_i = -\ln\left(\frac{1 - e^{-\beta_i g_i}}{1 + (g_i - 1)e^{-\beta_i g_i}}\right),$$

we recover

$$k\left(\mathbf{x}, \mathbf{x}'\right) = \prod_{i=1}^{n}\left(\frac{1 - e^{-\beta_i g_i}}{1 + (g_i - 1)e^{-\beta_i g_i}}\right)^{-\delta\left(x_i, x_i'\right)} \propto \prod_{i=1}^{n}\left(\frac{1 - e^{-\beta_i g_i}}{1 + (g_i - 1)e^{-\beta_i g_i}}\right)^{1 - \delta\left(x_i, x_i'\right)},$$

which is the ARD version of Equation 3.

For `COMBO`, we can rewrite Equation 5 as

$$k(\mathbf{x}, \mathbf{x}') = \prod_{i=1}^{n}\sum_{j=1}^{|\mathcal{X}_i|} e^{-\beta_i \lambda_j^i} f_j^i\left(x_i\right) f_j^i\left(x_i'\right) = \prod_{i=1}^{n} k_i\left(x_i, x_i'\right),$$

where

$$k_i\left(x_i, x_i'\right) = \sum_{j=1}^{|\mathcal{X}_i|} e^{-\beta_i \lambda_j^i} f_j^i\left(x_i\right) f_j^i\left(x_i'\right).$$

Now, Kondor and Lafferty (2002) show that, when the subgraphs $\mathcal{G}_i$ are complete, which is the case for the Hamming graph of Definition 2, $k_i\left(x_i, x_i'\right)$ becomes

$$k_i\left(x_i, x_i'\right) = \begin{cases} \frac{1 + (g_i - 1)e^{-\beta_i g_i}}{g_i} & \text{if } x_i = x_i' \\ \frac{1 - e^{-\beta_i g_i}}{g_i} & \text{if } x_i \neq x_i', \end{cases}$$

where $g_i = |\mathcal{X}_i|$ is the number of vertices in the complete subgraph $\mathcal{G}_i = (\mathcal{X}_i, \mathcal{E}_i)$. Using this simplified $k_i\left(x_i, x_i'\right)$, we obtain

$$k\left(\mathbf{x}, \mathbf{x}'\right) = \prod_{i=1}^{n}\left(\frac{1 + (g_i - 1)e^{-\beta_i g_i}}{g_i}\right)^{\delta\left(x_i, x_i'\right)}\left(\frac{1 - e^{-\beta_i g_i}}{g_i}\right)^{1 - \delta\left(x_i, x_i'\right)}$$

$$\propto \prod_{i=1}^{n}\left(\frac{1 - e^{-\beta_i g_i}}{1 + (g_i - 1)e^{-\beta_i g_i}}\right)^{1 - \delta\left(x_i, x_i'\right)},$$

which is the ARD version of Equation 3. $\qquad\square$

## A.2 Proof of Proposition 7

**Proposition 7.** *Let* $k : \mathbb{R}^n \times \mathbb{R}^n \to \mathbb{R}$ *be symmetric positive semi-definite and*

$$k(\mathbf{z}, \mathbf{z}') = \Bbbk\left(\|\mathbf{z} - \mathbf{z}'\|_2\right), \quad \text{with } \mathbf{z}, \mathbf{z}' \in \mathbb{R}^n,$$

*for some* $\Bbbk : \mathbb{R} \to \mathbb{R}$. *Then, the kernel*

$$k^*(\mathbf{x}, \mathbf{x}') = \Bbbk\left(\sqrt{h(\mathbf{x}, \mathbf{x}')}\right), \quad \text{with } \mathbf{x}, \mathbf{x}' \in \mathcal{X},$$

*is always symmetric positive semi-definite.*

*Proof.* Without loss of generality, assume $\mathbf{z}, \mathbf{z}'$ are the one-hot encoded versions of $\mathbf{x}, \mathbf{x}'$. Now, we have

$$k^*(\mathbf{x}, \mathbf{x}') = \Bbbk\left(\sqrt{h(\mathbf{x}, \mathbf{x}')}\right) = \Bbbk\left(\|\mathbf{z} - \mathbf{z}'\|_2 / \sqrt{2}\right) \propto k(\mathbf{z}, \mathbf{z}').$$

Since we assumed $k$ to be symmetric positive semi-definite, it follows that $k^*$ is also symmetric positive semi-definite. $\qquad\square$

## A.3 Proof of Theorem 9

We prove Theorem 9 by showing that (1) every $\Phi$-kernel is *isotropic*, (2) every isotropic kernel is a Hamming kernel and vice versa, (3) every Hamming kernel is a $\Phi$-kernel. First, we introduce the class of isotropic kernels.

**Definition 10.** A kernel $k : \mathcal{X} \times \mathcal{X} \to \mathbb{R}$ is called *isotropic* (with respect to the graph $\mathcal{G} = (\mathcal{X}, \mathcal{E})$) if and only if

$$k(\theta(\mathbf{x}), \theta(\mathbf{x}')) = k(\mathbf{x}, \mathbf{x}'), \qquad\qquad \mathbf{x}, \mathbf{x}' \in \mathcal{X}, \qquad\qquad \theta \in \mathrm{Aut}(\mathcal{G}),$$

where $\mathrm{Aut}(\mathcal{G})$ denotes the group of automorphisms of the graph $\mathcal{G}$. This group consists of maps $\theta : \mathcal{X} \to \mathcal{X}$ such that $(\theta(\mathbf{x}), \theta(\mathbf{x}')) \in \mathcal{E}$ is equivalent to $(\mathbf{x}, \mathbf{x}') \in \mathcal{E}$ for all $\mathbf{x}, \mathbf{x}' \in \mathcal{X}$.

It is easy to prove that every $\Phi$-kernel is isotropic. Moreover, this holds for general graphs $\mathcal{G}$ without restriction.

**Lemma 11.** *Let $k : \mathcal{X} \times \mathcal{X} \to \mathbb{R}$ be a $\Phi$-kernel in the sense of Definition 8, where $\mathcal{X}$ is the node set of an arbitrary graph $\mathcal{G} = (\mathcal{X}, \mathcal{E})$. Then, we have that $k$ is isotropic with respect to $\mathcal{G}$.*

*Proof.* Take $\theta \in \mathrm{Aut}(\mathcal{G})$. We prove $k(\theta(\mathbf{x}), \theta(\mathbf{x}')) = k(\mathbf{x}, \mathbf{x}')$. For convenience, we denote $|\mathcal{X}| = g$.

In this proof, it is more convenient to view the eigenfunctions $f_j$ as vectors $\boldsymbol{f}_j = (f_j(1), \dots, f_j(g))^\top$, the graph Laplacian $\Delta$ as its matrix $\boldsymbol{\Delta}$, and $\theta$, a permutation of the set $\{1, \dots, g\}$, as the permutation matrix $\boldsymbol{\Theta}$, such that $\boldsymbol{\Theta}(v_1, \dots, v_g)^\top = (v_{\theta(1)}, \dots, v_{\theta(g)})^\top$. By definition, $\boldsymbol{\Delta} = \mathbf{D} - \mathbf{A}$ where $\mathbf{A}$ is the adjacency matrix of $\mathcal{G}$ and $\mathbf{D}$ is the degree matrix, i.e. the diagonal matrix with $\mathbf{D}_{ii} = \sum_{j=1}^g \mathbf{A}_{ij}$.

In matrix form, the fact that $\theta \in \mathrm{Aut}(\mathcal{G})$ means $\boldsymbol{\Theta}^\top \mathbf{A} \boldsymbol{\Theta} = \mathbf{A}$, which is equivalent to $\mathbf{A}\boldsymbol{\Theta} = \boldsymbol{\Theta}\mathbf{A}$, because $\boldsymbol{\Theta}$ is an orthogonal matrix. Thus, $\boldsymbol{\Delta}\boldsymbol{\Theta} = \boldsymbol{\Theta}\boldsymbol{\Delta}$ and $\boldsymbol{\Delta}\boldsymbol{f}_j = \lambda_j \boldsymbol{f}_j$ implies $\boldsymbol{\Delta}\boldsymbol{\Theta}\boldsymbol{f}_j = \lambda_j \boldsymbol{\Theta}\boldsymbol{f}_j$. This, together with orthogonality of $\boldsymbol{\Theta}$, implies that $\{\boldsymbol{\Theta}\boldsymbol{f}_j\}_{j=1}^g$ is an orthonormal basis of eigenvectors corresponding to eigenvalues $\lambda_j$, same as $\{\boldsymbol{f}_j\}_{j=1}^g$. What is the same, $\{f_j \circ \theta\}_{j=1}^g$ is an orthonormal basis of eigenfunctions corresponding to eigenvalues $\lambda_j$. Since $\Phi$-kernels do not depend on the choice of an orthonormal basis of eigenfunctions, we have

$$k(\theta(\mathbf{x}), \theta(\mathbf{x}')) = \sum_{j=1}^g \Phi(\lambda_j) f_j(\theta(\mathbf{x})) f_j(\theta(\mathbf{x}')) = k(\mathbf{x}, \mathbf{x}'),$$

proving that $k$ is isotropic. $\qquad\square$

*Remark.* The converse of Lemma 11 is not, in general, true. Intuitively, when the group $\mathrm{Aut}(\mathcal{G})$ is small, it becomes easier to be isotropic, and the class of isotropic kernels can become larger than the class of $\Phi$-kernels. For example, in the extreme case when $\mathrm{Aut}(\mathcal{G})$ is trivial, all kernels are isotropic, but not all kernels are $\Phi$-kernels.

To continue working with isotropic kernels, we need to characterize the automorphisms of $\mathcal{G}$. Let us start with the simplest case. If $\mathcal{G}$ is a complete graph with $|\mathcal{X}| = g$ nodes, then any bijection $\theta : \mathcal{X} \to \mathcal{X}$ is an automorphism. All such bijections are permutations of a $g$-element set and form the symmetric group $\mathrm{S}_g$, therefore $\mathrm{Aut}(\mathcal{G}) = \mathrm{S}_g$.

Now, let us consider the Hamming graphs of Definition 2. Specifically, we consider $\mathcal{G} = \square_{i=1}^n \mathcal{G}_i$ where $\mathcal{G}_i = (\mathcal{X}_i, \mathcal{E}_i)$ are complete graphs and assume all factors to be of the same size, i.e. $|\mathcal{X}_1| = |\mathcal{X}_2| = \dots = |\mathcal{X}_n|$. In this case, we can assume $\mathcal{G}_1 = \dots = \mathcal{G}_n$ without loss of generality.

**Lemma 12.** *Consider $\mathcal{G} = \square_{i=1}^{n} \mathcal{G}_i$ with $\mathcal{G}_1 = \ldots = \mathcal{G}_n$ complete graphs having $g$ nodes. If $\theta \in \mathrm{Aut}(\mathcal{G})$, then*

$$\theta(x_1, \ldots, x_n) = (\theta_1(x_{\sigma(1)}), \ldots, \theta_n(x_{\sigma(n)})),$$

*where $\sigma \in \mathrm{S}_n$ is a permutation of the set $\{1, \ldots, n\}$ and $\theta_i \in \mathrm{Aut}(\mathcal{G}_i) = \mathrm{S}_g$. That is, $\mathrm{Aut}(\mathcal{G}) = S_g^n \rtimes S_n$, where $\rtimes$ denotes the semidirect product of groups (see, e.g., Robinson (2003) for the definition of a semidirect product).*

*Proof.* Follows from Imrich et al. (2000, Theorem 4.15). $\qquad\square$

With this, we can connect Hamming kernels to the isotropic kernels.

**Lemma 13.** *Let $\mathcal{G} = (\mathcal{X}, \mathcal{E})$ be a Hamming graph from Definition 2, with $|\mathcal{X}_1| = |\mathcal{X}_2| = \ldots = |\mathcal{X}_n|$, then $k : \mathcal{X} \times \mathcal{X} \to \mathbb{R}$ is isotropic (with respect to $\mathcal{G}$) if and only if $k$ is a function of Hamming distance.*

*Proof.* To simplify notation, we assume, without loss of generality, $\mathcal{X}_i = \{0, 1, \ldots, g - 1\}$. Take $\mathbf{x}, \mathbf{x}' \in \mathcal{X}$. Consider $\theta \in \mathrm{Aut}(\mathcal{G})$ with $\theta(\mathbf{x}) = (\theta_1(x_1), \ldots, \theta_n(x_n))$. Take $\theta_i$ with $\theta_i(x_i') = 0$ and $\theta_i(x_i) = \mathbb{1}_{x_i \neq x_i'}$. This constitutes a valid automorphism of $\mathcal{G}$. Thus, we have

$$k(\mathbf{x}, \mathbf{x}') = k(\theta(\mathbf{x}), \theta(\mathbf{x})) = k(\widetilde{\mathbf{x}}, \mathbf{0}),$$

where $\widetilde{\mathbf{x}} = (\widetilde{x}_1, \ldots, \widetilde{x}_n)$ and $\widetilde{x}_i = 0$ if $x_i = x_i'$ or $\widetilde{x}_i = 1$ otherwise. We can further consider $\widetilde{\theta} \in \mathrm{Aut}(\mathcal{G})$ such that $\widetilde{\theta}(\mathbf{x}) = (x_{\sigma(1)}, \ldots, x_{\sigma(n)})$ where $\sigma$ is such that $\widetilde{\theta}(\widetilde{\mathbf{x}})$ is of form $(1, 1, \ldots, 1, 0, 0, \ldots, 0)$. If $\hat{\mathbf{x}} = \sigma(\widetilde{\mathbf{x}})$, then

$$k(\mathbf{x}, \mathbf{x}') = k(\widetilde{\mathbf{x}}, \mathbf{0}) = k(\hat{\mathbf{x}}, \mathbf{0}).$$

It follows that $k$ depends only on the number of indices $i$ for which $x_i = x_i'$, but not on particular values of $x_i, x_i'$ and not on the particular indices $i$. That is, $k$ only depends on the Hamming distance between $\mathbf{x}$ and $\mathbf{x}'$.

Conversely, assume $k$ is a function of Hamming distance, or rather of its square root, as stated in Definition 6. Take $\theta \in \mathrm{Aut}(\mathcal{G})$. We prove $k(\theta(\mathbf{x}), \theta(\mathbf{x}')) = k(\mathbf{x}, \mathbf{x}')$. Here, the key observation is that $h(\theta(\mathbf{x}), \theta(\mathbf{x}')) = h(\mathbf{x}, \mathbf{x}')$, which is obvious from the form of $\theta$. This immediately implies

$$k(\theta(\mathbf{x}), \theta(\mathbf{x}')) = \Bbbk\left(\sqrt{h(\theta(\mathbf{x}), \theta(\mathbf{x}'))}\right) = \Bbbk\left(\sqrt{h(\mathbf{x}, \mathbf{x}')}\right) = k(\mathbf{x}, \mathbf{x}')$$

and proves the claim. $\qquad\square$

Finally, we prove that a Hamming kernel is a $\Phi$-kernel.

**Lemma 14.** *If $k : \mathcal{X} \times \mathcal{X} \to \mathbb{R}$ is a kernel which is a function of Hamming distance, then $k$ is a $\Phi$-kernel.*

*Proof.* Let us start with the simplest case, when $\mathcal{G} = \mathcal{G}_1$ is a complete graph with $\mathcal{X} = \{0, \ldots, g-1\}$. For such $\mathcal{G}$, the first eigenfunction $f_1$ of the Laplacian is the constant function $f_1(x) \equiv 1$, which correspond to eigenvalue $0$. Other eigenfunctions $f_2, \ldots, f_g$ are an arbitrary orthonormal basis in the linear space of functions that sum up to $0$: $\{f : \mathcal{X} \to \mathbb{R} : \sum_{j=0}^{g-1} f(j) = 0\}$, which correspond to eigenvalue $g$.

On the other hand, in this case we have $k(x, x') = \alpha \mathbb{1}_{x=x'} + \beta \mathbb{1}_{x \neq x'}$ as $k$ may only take two distinct values, depending on whether $x$ is equal to $x'$ or not. We prove that $k$ is a $\Phi$-kernel using Mercer's theorem. For this, we check that the eigenfunctions of the Laplacian $f_j$ are also eigenfunctions of the covariance operator $(kf)(x) = \sum_{x'=0}^{g-1} k(x, x') f(x')$. Furthermore, we verify that if two eigenfunctions $f_i, f_j$ correspond to the same eigenvalue $\lambda$ of the Laplacian, then they correspond to the same eigenvalue $\lambda'$—note that $\lambda'$ can differ from $\lambda$—of the covariance operator $k$, i.e that $\Delta f_i = \lambda f_i$ and $\Delta f_j = \lambda f_j$ implies $k f_i = \lambda' f_i$, $k f_j = \lambda' f_j$.

Let us show that $f_1$ is an eigenfunction of $k$. Write

$$\sum_{x'=0}^{g-1} k(x, x') f_1(x') = \sum_{x'=0}^{g-1} k(x, x') = \alpha + (g-1)\beta = (\alpha + (g-1)\beta) f_1(x).$$

Similarly, we can show that $f_2, \ldots, f_g$ are eigenfunctions of $k$. Take $j \geq 2$ and write

$$\sum_{x'=0}^{g-1} k(x, x') f_j(x') = \alpha f_j(x) + \beta \overbrace{\sum_{\substack{0 \leq x' \leq g-1 \\ x' \neq x}} f_j(x')}^{-f_j(x)} = (\alpha - \beta) f_j(x).$$

This also shows that the eigenvalue corresponding to $f_j$ with $j \geq 2$ is the same and completes the proof for the simple case when $\mathcal{G} = \mathcal{G}_1$.

The next step is to consider a Hamming graph $\mathcal{G} = \square_{i=1}^n \mathcal{G}_i$ where $\mathcal{G}_i = (\mathcal{X}_i, \mathcal{E}_i)$ are complete graphs with $|\mathcal{X}_1| = |\mathcal{X}_2| = \ldots = |\mathcal{X}_n| = g$. Without loss of generality, we assume $\mathcal{G}_1 = \mathcal{G}_2 = \ldots = \mathcal{G}_n$. Let $f_j$ denote the eigenfunctions of the graph Laplacian on the factors $\mathcal{G}_i$, such that $f_1, \ldots, f_g$ form an orthonormal basis, exactly as in the simple case considered above. Then, an orthonormal basis consisting of eigenfunctions of the graph Laplacian on $\mathcal{G}$ can be obtained by considering all possible products $f_{j_1, \ldots, j_n}(x_1, \ldots, x_n) = f_{j_1}(x_1) \ldots f_{j_n}(x_n)$:

$$\Delta f_{j_1, \ldots, j_n} = \underbrace{(\lambda_{j_1} + \ldots + \lambda_{j_n})}_{\lambda_{j_1, \ldots, j_n}} f_{j_1, \ldots, j_n}.$$

Since $\lambda_1 = 0$ and $\lambda_2 = \ldots = \lambda_g = g$, we have $\lambda_{i_1, \ldots, i_n} = \lambda_{j_1, \ldots, j_n}$ if and only if the tuples $(i_1, \ldots, i_n)$ and $(j_1, \ldots, j_n)$ have the same number of zeros.

Under our assumptions, $k$ must be of form $k(\mathbf{x}, \mathbf{x}') = \kappa(\mathbb{1}_{x_1 = x'_1}, \mathbb{1}_{x_2 = x'_2}, \ldots, \mathbb{1}_{x_n = x'_n})$ for some $\kappa : \{0,1\}^n \to \mathbb{R}$. Let us prove that $f_{j_1, \ldots, j_n}$ are also eigenfunctions of the operator $(kf)(\mathbf{x}) = \sum_{\mathbf{x}' \in \mathcal{X}} k(\mathbf{x}, \mathbf{x}') f(\mathbf{x}')$ and that $k f_{j_1, \ldots, j_n} = \lambda'_{j_1, \ldots, j_n} f_{j_1, \ldots, j_n}$ where $\lambda'_{j_1, \ldots, j_n}$ only depends on the number of zeros in the tuple $(j_1, \ldots, j_n)$. By Mercer's theorem, this implies that $k$ is a $\Phi$-kernel.

Without loss of generality, assume $f(\mathbf{x}) = f_{j_1}(x_1) f_{j_2}(x_2) \ldots f_{j_l}(x_l)$ for some $l \leq n$, where $j_1, j_2, \ldots, j_l \geq 2$, i.e. $f$ is a product of $l$ non-constant eigenfunctions and $n - l$ constant ones, ordered such that the non-constant ones come first. Denote $\mathbf{x}_{i:} = (x_i, \ldots, x_n)$, $\mathcal{X}_{i:} = \mathcal{X}_i \times \ldots \times \mathcal{X}_n$, and $f_{i:}(\mathbf{x}_{i:}) = f_{j_i}(x_i) \ldots f_{j_l}(x_l)$. Write

$$(kf)(\mathbf{x}) = \sum_{\mathbf{x}' \in \mathcal{X}} k(\mathbf{x}, \mathbf{x}') f(\mathbf{x}') = \sum_{\mathbf{x}' \in \mathcal{X}} \kappa(\mathbb{1}_{x_1 = x'_1}, \mathbb{1}_{x_2 = x'_2}, \ldots, \mathbb{1}_{x_n = x'_n}) f(\mathbf{x}')$$

$$= \sum_{\mathbf{x}'_{2:} \in \mathcal{X}_{2:}} \left( \sum_{x'_1 = 0}^{g-1} \kappa(\mathbb{1}_{x_1 = x'_1}, \mathbb{1}_{x_2 = x'_2}, \ldots, \mathbb{1}_{x_n = x'_n}) f_{j_1}(x'_1) \right) f_{2:}(\mathbf{x}'_{2:}).$$

Now, using the fact that $\sum_{\substack{0 \leq x'_1 \leq g-1 \\ x'_1 \neq x_1}} f_{j_1}(x'_1) = -f_{j_1}(x_1)$, continue

$$= f_{j_1}(x_1) \sum_{\mathbf{x}'_{2:} \in \mathcal{X}_{2:}} \left( \kappa(1, \mathbb{1}_{x_2 = x'_2}, \ldots, \mathbb{1}_{x_n = x'_n}) - \kappa(0, \mathbb{1}_{x_2 = x'_2}, \ldots, \mathbb{1}_{x_n = x'_n}) \right) f_{2:}(\mathbf{x}'_{2:})$$

$$= f_{j_1}(x_1) \sum_{\mathbf{x}'_{3:} \in \mathcal{X}_{3:}} \left( \sum_{x'_2 = 0}^{g_2 - 1} \kappa(1, \mathbb{1}_{x_2 = x'_2}, \ldots, \mathbb{1}_{x_n = x'_n}) f_{j_2}(x'_2) \right) f_{3:}(\mathbf{x}'_{3:})$$

$$- f_{j_1}(x_1) \sum_{\mathbf{x}'_{3:} \in \mathcal{X}_{3:}} \left( \sum_{x'_2 = 0}^{g_2 - 1} \kappa(0, \mathbb{1}_{x_2 = x'_2}, \ldots, \mathbb{1}_{x_n = x'_n}) f_{j_2}(x'_2) \right) f_{3:}(\mathbf{x}'_{3:})$$

$$= f_{j_1}(x_1) f_{j_2}(x_2) \sum_{\mathbf{x}'_{3:} \in \mathcal{X}_{3:}} \left( \kappa(1, 1, \ldots, \mathbb{1}_{x_n = x'_n}) - \kappa(1, 0, \ldots, \mathbb{1}_{x_n = x'_n}) \right) f_{3:}(\mathbf{x}'_{3:})$$

$$- f_{j_1}(x_1) f_{j_2}(x_2) \sum_{\mathbf{x}'_{3:} \in \mathcal{X}_{3:}} \left( \kappa(0, 1, \ldots, \mathbb{1}_{x_n = x'_n}) - \kappa(0, 0, \ldots, \mathbb{1}_{x_n = x'_n}) \right) f_{3:}(\mathbf{x}'_{3:}).$$

Repeating the same steps multiple times, we arrive at

$$= f_{j_1}(x_1) f_{j_2}(x_2) \ldots f_{j_l}(x_l) \underbrace{\sum_{\mathbf{a} \in \{0,1\}^l} (-1)^{1 + \sum_i a_i} \sum_{\mathbf{x}'_{l+1:} \in \mathcal{X}_{l+1:}} \kappa(a_1, \ldots, a_l, \mathbb{1}_{x_{l+1} = x'_{l+1}}, \mathbb{1}_{x_n = x'_n})}_{\text{a constant, denote it by } \lambda}$$

It follows that $(kf)(\mathbf{x}) = \lambda f(\mathbf{x})$, where $\lambda$ is the eigenvalue, equal to the alternating sum of above. Importantly, since $k$ is a function of Hamming distance, $\kappa$ only depend on the number of occurrences of 1 in its arguments, thus $\lambda$ only depends on $l$, or, equivalently, on $n - l$, which is precisely the number of zeros in the tuple of indexes. This proves the claim. $\qquad\square$

**Theorem 9.** *Let $\mathcal{G} = (\mathcal{X}, \mathcal{E})$ be a Hamming graph with $|\mathcal{X}_1| = \cdots = |\mathcal{X}_n|$. Then, the class of Hamming kernels on $\mathcal{X}$ coincides with the class of $\Phi$-kernels on $\mathcal{G}$.*

*Proof.* By Lemma 14, every Hamming kernel is a $\Phi$-kernel. By the combination of Lemma 11 and Lemma 13, every $\Phi$-kernel is a Hamming kernel. $\qquad\square$

We now show that this is not, in general, the case for Hamming graphs for which $|\mathcal{X}_1|, |\mathcal{X}_2|, \ldots, |\mathcal{X}_n|$ can differ.

*Example.* Consider $\mathcal{X}_1 = \mathcal{X}_2 = \{0, 1\}$ and $\mathcal{X}_3 = \{0, 1, 2, 3\}$, the respective complete graphs $\mathcal{G}_i = (\mathcal{X}_i, \mathcal{E}_i)$, and their product $\mathcal{G} = \square_{i=1}^3 \mathcal{G}_i$ with $\mathcal{G} = (\mathcal{X}, \mathcal{E})$, $\mathcal{X} = \mathcal{X}_1 \times \mathcal{X}_2 \times \mathcal{X}_3$.

Since $\mathcal{G}$ is a Cartesian product of $\mathcal{G}_i$, every eigenvalue $\lambda$ of the graph Laplacian on $\mathcal{G}$ is of form $\lambda = \lambda_1 + \lambda_2 + \lambda_3$, where $\lambda_i$ is an eigenvalue of the graph Laplacian on $\mathcal{G}_i$. On $\mathcal{G}_1 = \mathcal{G}_2$, there are two distinct eigenvalues: 0 and 2. The distinct eigenvalues on $\mathcal{G}_3$ are 0 and 4. This means that there are 5 distinct eigenvalues on $\mathcal{G}$:

$$0 = 0 + 0 + 0,$$
$$2 = 2 + 0 + 0 = 0 + 2 + 0,$$
$$4 = 2 + 2 + 0 = 0 + 0 + 4,$$
$$6 = 2 + 0 + 4 = 0 + 2 + 4,$$
$$8 = 2 + 2 + 4.$$

It follows that the eigenvalues of all $\Phi$-kernels are of form $\Phi(0), \Phi(2), \Phi(4), \Phi(6), \Phi(8)$, which implies that a $\Phi$-kernel on $\mathcal{G}$ can have no more than 5 distinct eigenvalues. We will show an example of a kernel $k : \mathcal{X} \times \mathcal{X} \to \mathbb{R}$ which is a function of the Hamming distance on $\mathcal{G}$, and which has 6 distinct eigenvalues. This will prove that the class of $\Phi$-kernels on $\mathcal{G}$ is strictly smaller than the class of Hamming kernels.

Consider the kernel $k : \mathcal{X} \times \mathcal{X} \to \mathbb{R}$ defined by

$$k(\mathbf{x}, \mathbf{x}') = \begin{cases} 10, & h(\mathbf{x}, \mathbf{x}') = 0, \\ 6, & h(\mathbf{x}, \mathbf{x}') = 1, \\ 4, & h(\mathbf{x}, \mathbf{x}') = 2, \\ 3, & h(\mathbf{x}, \mathbf{x}') = 3, \end{cases}$$

where $h(\cdot, \cdot')$ is the Hamming distance on $\mathcal{G}$. Consider the ordering of all possible inputs in the following way:

$$(0, 0, 0), \ (0, 0, 1), \ (0, 0, 2), \ (0, 0, 3), \ (0, 1, 0), \ (0, 1, 1), \ (0, 1, 2), \ (0, 1, 3),$$
$$(1, 0, 0), \ (1, 0, 1), \ (1, 0, 2), \ (1, 0, 3), \ (1, 1, 0), \ (1, 1, 1), \ (1, 1, 2), \ (1, 1, 3).$$

Then $k$ can be represented as a matrix of size $|\mathcal{X}| \times |\mathcal{X}| = 16 \times 16$. This matrix is

$$\begin{bmatrix}
10 & 6 & 6 & 6 & 6 & 4 & 4 & 4 & 6 & 4 & 4 & 4 & 4 & 3 & 3 & 3 \\
6 & 10 & 6 & 6 & 4 & 6 & 4 & 4 & 4 & 6 & 4 & 4 & 3 & 4 & 3 & 3 \\
6 & 6 & 10 & 6 & 4 & 4 & 6 & 4 & 4 & 4 & 6 & 4 & 3 & 3 & 4 & 3 \\
6 & 6 & 6 & 10 & 4 & 4 & 4 & 6 & 4 & 4 & 4 & 6 & 3 & 3 & 3 & 4 \\
6 & 4 & 4 & 4 & 10 & 6 & 6 & 6 & 4 & 3 & 3 & 3 & 6 & 4 & 4 & 4 \\
4 & 6 & 4 & 4 & 6 & 10 & 6 & 6 & 3 & 4 & 3 & 3 & 4 & 6 & 4 & 4 \\
4 & 4 & 6 & 4 & 6 & 6 & 10 & 6 & 3 & 3 & 4 & 3 & 4 & 4 & 6 & 4 \\
4 & 4 & 4 & 6 & 6 & 6 & 6 & 10 & 3 & 3 & 3 & 4 & 4 & 4 & 4 & 6 \\
6 & 4 & 4 & 4 & 4 & 3 & 3 & 3 & 10 & 6 & 6 & 6 & 6 & 4 & 4 & 4 \\
4 & 6 & 4 & 4 & 3 & 4 & 3 & 3 & 6 & 10 & 6 & 6 & 4 & 6 & 4 & 4 \\
4 & 4 & 6 & 4 & 3 & 3 & 4 & 3 & 6 & 6 & 10 & 6 & 4 & 4 & 6 & 4 \\
4 & 4 & 4 & 6 & 3 & 3 & 3 & 4 & 6 & 6 & 6 & 10 & 4 & 4 & 4 & 6 \\
4 & 3 & 3 & 3 & 6 & 4 & 4 & 4 & 6 & 4 & 4 & 4 & 10 & 6 & 6 & 6 \\
3 & 4 & 3 & 3 & 4 & 6 & 4 & 4 & 4 & 6 & 4 & 4 & 6 & 10 & 6 & 6 \\
3 & 3 & 4 & 3 & 4 & 4 & 6 & 4 & 4 & 4 & 6 & 4 & 6 & 6 & 10 & 6 \\
3 & 3 & 3 & 4 & 4 & 4 & 4 & 6 & 4 & 4 & 4 & 6 & 6 & 6 & 6 & 10
\end{bmatrix}$$

Its eigenvalues, repeated according to multiplicities, are $77, 15, 15, 9, 9, 9, 5, 3, 3, 3, 3, 3, 3, 1, 1, 1$. There are 6 unique eigenvalues: $77, 15, 9, 5, 3, 1$, showing that the class of Hamming kernels is larger than the class of $\Phi$-kernels on $\mathcal{G}$.

### A.4 Proof that BODi is not a (variation of a) heat kernel

As outlined in Definition 6: a kernel is a Hamming Kernel (HK) if and only if it depends on $\mathbf{x}, \mathbf{x}' \in \mathcal{X}$ through the square root of the Hamming distance $h(\mathbf{x}, \mathbf{x}')$. That is,

$$k_{\text{HK}}(\mathbf{x}, \mathbf{x}') := f\left(\sqrt{h(\mathbf{x}, \mathbf{x}')}\right),$$

for some $f : \mathbb{Z}_0^+ \to \mathbb{R}$.

In contrast, the BODi kernel depends on $\mathbf{x}, \mathbf{x}' \in \mathcal{X}$ through the term

$$k_{\text{BODi}}(\mathbf{x}, \mathbf{x}') := g\left(\|\phi_{\mathbf{A}}(\mathbf{x}) - \phi_{\mathbf{A}}(\mathbf{x}')\|_2^2\right),$$

with

$$g(d) := \left(1 + \frac{\sqrt{5}d}{\ell} + \frac{5d^2}{3\ell^2}\right) \exp\left(-\frac{\sqrt{5}d}{\ell}\right).$$

Here, $g(\cdot)$ is the Matérn-5/2 function, where we note that $g(\cdot)$ constitutes a bijective function. Additionally, $\phi_{\mathbf{A}}(\cdot)$ denotes the mapping towards the HED embedding (see Section 4 of Deshwal et al., 2023), namely:

$$[\phi_{\mathbf{A}}(\mathbf{x})]_i := h(\mathbf{a}_i, \mathbf{x}),$$

with $\mathbf{a}_i \in \mathbf{A}$ one of the $M$ anchor points in dictionary $\mathbf{A}$.

For BODi to be a Hamming kernel, there thus needs to exist a function, say $u(\cdot)$, such that

$$u\left(\sqrt{h(\mathbf{x}, \mathbf{x}')}\right) = \|\phi_{\mathbf{A}}(\mathbf{x}) - \phi_{\mathbf{A}}(\mathbf{x}')\|_2^2.$$

If this function $u(\cdot)$ exists, we can define $f := g \circ u$, obtaining

$$k_{\text{BODi}}(\mathbf{x}, \mathbf{x}') = g\left(\|\phi_{\mathbf{A}}(\mathbf{x}) - \phi_{\mathbf{A}}(\mathbf{x}')\|_2^2\right) = g\left(u\left(\sqrt{h(\mathbf{x}, \mathbf{x}')}\right)\right) = f\left(\sqrt{h(\mathbf{x}, \mathbf{x}')}\right) = k_{\text{HK}}(\mathbf{x}, \mathbf{x}').$$

However, as becomes clear by looking at the mapping $\phi_{\mathbf{A}}(\cdot)$, there does not exist a function $u(\cdot)$ satisfying the above claim. We provide a small counter-example with $M = 1$ below.

In scenario A, we have:

$$\mathbf{x} = [1, 1, 1],\ \mathbf{x}' = [1, 1, 2] \text{ and } \mathbf{a}_1 = [1, 1, 1],$$

which gives us

$$\sqrt{h(\mathbf{x}, \mathbf{x}')} = 1 \text{ and } \|\phi_{\mathbf{A}}(\mathbf{x}) - \phi_{\mathbf{A}}(\mathbf{x}')\|_2^2 = 1.$$

In scenario B, we have:

$$\mathbf{x} = [1, 1, 2],\ \mathbf{x}' = [1, 1, 3] \text{ and } \mathbf{a}_1 = [1, 1, 1],$$

which gives us

$$\sqrt{h(\mathbf{x}, \mathbf{x}')} = 1 \text{ and } \|\phi_{\mathbf{A}}(\mathbf{x}) - \phi_{\mathbf{A}}(\mathbf{x}')\|_2^2 = 0.$$

For $u(\cdot)$ to work in both scenarios, we would need to have:

$$u(1) = 1 \text{ and } u(1) = 0,$$

which is clearly not possible. As a result, we conclude that BODi is not a Hamming kernel, and therefore also not a heat kernel (or a direct generalization thereof).

In fact, using the above proof structure, we can show an even stronger result: BODi is not part of an (even broader) general class of kernels, which includes not only Hamming- and graph-based kernels, but also additive kernels such as CoCaBO and RDUCB (see Appendix D.2). With this, we feel confident in our answer that BODi cannot be seen as a heat-kernel variation in some sense, and its HED kernel is thus fundamentally different from the kernels grouped by our unifying framework.

## B    Ordinal variables in combinatorial Bayesian optimization

Combinatorial Bayesian optimization is often used as umbrella term to denote a variety of discrete search spaces, resulting in potential misunderstandings. In this paper, we argue combinatorial BO is centered on three main types of variables: categorical, ordinal, and "discrete-quantitative". *Categorical* variables can take elements from a discrete, finite, and unordered set, where the distances between the different elements of the set are unknown. For *ordinal* variables, the same definition applies, except the set is now ordered (but distances are still undefined). In contrast, *discrete-quantitative* variables—which are often mistakenly referred to as ordinal variables—not only possess an order between the different elements, but also a specific distance.

As a general example, we consider the number of layers in a neural network. Here, although the variable is discrete, there is a clear distance between the different categories (namely the Euclidean distance). Therefore, according to our definition, this is not an ordinal variable, but rather a discrete-quantitative variable. As can be imagined, truly ordinal variables are rare, and are in fact not considered in any of the kernels/methods that we compare against in this paper (to the best of our knowledge).

For discrete-quantitative variables, we believe the associated kernels are rather straightforward: use any (continuous) isotropic kernel, along with an appropriate distance function (usually Euclidean). As such, these kernels can simply be seen as continuous kernels, which is why we did not explicitly address them in our unifying framework. For ordinal variables (although rarely considered in the field), we suggest replacing the complete graphs $\mathcal{G}_i$ from Definition 2 by path graphs, where nodes are only connected to the category right above or below in the ordered set $\mathcal{X}$. This approach is similar to what is proposed in Oh et al. (2019), and allows us to connect ordinal variables with our unifying framework centered on heat kernels.

Unfortunately, the above connection is also where the analysis stops. Since path graphs do not contain the same type of symmetries as complete graphs, the eigenvalues of the corresponding Laplacian do not simplify as much as for categorical variables. Therefore, we do not obtain a simple closed-form solution. In other words, Equations 2 or 5 cannot be simplified to something like Equations 3 or 4.

## C    Deriving heat kernels from general combinatorial kernels

In Appendix C.1, we define the most general combinatorial kernel and show that this leads to an explosion in degrees of freedom. In Appendix C.2, we demonstrate that applying two simple assumptions significantly reduces the degrees of freedom associated with the kernel. In Appendix C.3, we put both pieces together and arrive at Equation 3, therefore deriving heat kernels in a clear and systematic way.

### C.1    Defining general combinatorial kernels

For a finite input space $\mathcal{X}$, there are only a finite amount of pairs $(\mathbf{x}, \mathbf{x}') \in \mathcal{X} \times \mathcal{X}$ at which a kernel function $k(\cdot, \cdot)$ can be evaluated. As such, we can represent this function $k(\cdot, \cdot)$ as an $|\mathcal{X}| \times |\mathcal{X}|$ matrix $\mathbf{K}$, with rows and columns indexed by the elements of $\mathcal{X}$ and related to the kernel function by $K_{\mathbf{x}, \mathbf{x}'} = k(\mathbf{x}, \mathbf{x}')$ (Kondor and Lafferty, 2002). From this, it becomes clear that defining a combinatorial kernel $k$ amounts to defining a symmetric positive semi-definite matrix $\mathbf{K}$.

To define the most general combinatorial kernel $k$, we can simply populate $\mathbf{K}$ with hyperparameters and update those by maximizing the marginal log likelihood. By doing so, any pair of points $(\mathbf{x}, \mathbf{x}') \in \mathcal{X} \times \mathcal{X}$ can be assigned any covariance $k(\mathbf{x}, \mathbf{x}')$, resulting in the most expressive kernel $k$. To ensure symmetry and positive semi-definiteness, we can parameterize Cholesky factor $\mathbf{C}$ of $\mathbf{K} = \mathbf{C}\mathbf{C}^{\top}$, resulting in $|\mathcal{X}| \left( |\mathcal{X}| + 1 \right) / 2$ hyperparameters.

Clearly, for most combinatorial input spaces of interest, this number of hyperparameters far exceeds the number of datapoints typically considered in BO tasks. To significantly reduce the number of hyperparameters, we can impose two simple assumptions on the parameterization of $\mathbf{K}$: product decomposition and compound symmetry.

## C.2 Reducing degrees of freedom using simple assumptions

Instead of parameterizing $\mathbf{K}$ using the Cholesky decomposition, we can instead define $\mathbf{K}$ as being a Kronecker product of smaller matrices $\mathbf{K}_i$, similarly to how the standard (continuous) RBF kernel $k(\mathbf{x}, \mathbf{x}')$ can be seen as a product of one-dimensional kernels $k_i(x_i, x_i')$, each acting on its own variable.

**Assumption 15** (Product Decomposition). *Given a combinatorial input space $\mathcal{X} = \times_{i=1}^{n} \mathcal{X}_i$, the associated kernel matrix $\mathbf{K}$ can be written as*

$$\mathbf{K} = \bigotimes_{i=1}^{n} \mathbf{K}_i,$$

*where $\bigotimes$ is the Kronecker product and $\mathbf{K}_i$ are symmetric positive semi-definite matrices of size $|\mathcal{X}_i| \times |\mathcal{X}_i|$.*

By imposing Assumption 15, we reduce the degrees of freedom from $\mathcal{O}\left(|\mathcal{X}|^2\right)$ down to $\mathcal{O}\left(\sum_{i=1}^{n} |\mathcal{X}_i|^2\right)$. If we see $\mathbf{x} \in \mathcal{X}$ as an $n$-dimensional vector of categorical variables, then Assumption 15 minimizes the explosion in the number of hyperparameters due to the increase in the number of variables, but not due to the increase in the number of categories per variable. For the latter, we can impose a certain structure on each $\mathbf{K}_i$.

**Assumption 16** (Compound Symmetry). *For a kernel matrix $\mathbf{K}$ satisfying Assumption 15, we have*

$$\mathbf{K}_i\left[x, x'\right] = \left\{ \begin{array}{ll} v_i & \text{if } x = x' \\ c_i & \text{if } x \neq x' \end{array} \right. , \quad c_i/v_i \in \left(-1/\left(|\mathcal{X}_i| - 1\right), 1\right) , \tag{7}$$

*where $v_i$ and $c_i$ are hyperparameters.*

Each matrix $\mathbf{K}_i$ satisfying Equation 7 is positive semi-definite (Roustant et al., 2020).

By imposing Assumption 16, we obtain matrices $\mathbf{K}_i$ with only two different values: diagonals $v_i$, namely the variances, and off-diagonals $c_i$, namely the covariances. In essence, for each variable $x_i$, the covariances between the different categories are now considered equal, no matter which two categories are getting compared. By imposing both assumptions, we now have $2n$ hyperparameters. This number can be reduced further to $n + 1$ hyperparameters, as will be shown in the next section.

## C.3 Deriving heat kernels

In the previous two sections, we have represented $k(\cdot, \cdot)$ in terms of matrices, so as to easily represent the number of hyperparameters involved. In this section, we return to the function view and, after applying Assumptions 15 and 16, we obtain

$$k(\mathbf{x}, \mathbf{x}') = \prod_{i=1}^{n} k_i(x_i, x_i'), \quad \text{with } k_i(x_i, x_i') = \left\{ \begin{array}{ll} v_i & \text{if } x_i = x_i' \\ c_i & \text{if } x_i \neq x_i' \end{array} \right.$$

and $c_i/v_i \in \left(-1/\left(|\mathcal{X}_i| - 1\right), 1\right)$. Now, putting both assumptions together gives us

$$k(\mathbf{x}, \mathbf{x}') = \prod_{i=1}^{n} k_i(x_i, x_i') = \prod_{i=1}^{n} v_i^{\delta(x_i, x_i')} c_i^{1-\delta(x_i, x_i')} \propto \prod_{i=1}^{n} \left[\frac{c_i}{v_i}\right]^{1-\delta(x_i, x_i')} = \prod_{i=1}^{n} \rho_i^{1-\delta(x_i, x_i')}, \tag{8}$$

where $\delta(\cdot, \cdot)$ is the Kronecker delta function and $\rho_i \in \left(-1/\left(|\mathcal{X}_i| - 1\right), 1\right)$.

From Equation 8, it becomes clear that only $n + 1$ hyperparameters are needed, where the additional hyperparameter comes from the fact that we now need to explicitly add a signal variance $\sigma^2$. Furthermore, choosing

$$\rho_i = \frac{1 - e^{-\beta_i g_i}}{1 + (g_i - 1)e^{-\beta_i g_i}},$$

we get

$$k(\mathbf{x}, \mathbf{x}') = \prod_{i=1}^{n} \rho_i^{1-\delta(x_i, x_i')} = \prod_{i=1}^{n} \left[\frac{1 - e^{-\beta_i g_i}}{1 + (g_i - 1)e^{-\beta_i g_i}}\right]^{1-\delta(x_i, x_i')},$$

which is the ARD version of the heat kernel expressed in Equation 3. Similar to Appendix A, $\beta_i \geq 0$ are hyperparameters and $g_i = |\mathcal{X}_i|$, leading to

$$\rho_i = f(\beta_i) = \frac{1 - e^{-\beta_i g_i}}{1 + (g_i - 1)e^{-\beta_i g_i}} \in (-1/(|\mathcal{X}_i| - 1), 1),$$

therefore ensuring the base kernels $k(x_i, x_i')$ are still positive semi-definite.

## D    Extending heat kernels: incorporating invariances and additive structures

In Section 3, we have presented a unifying framework describing general classes of combinatorial kernels. In this appendix, we show that these classes of kernels can easily be modified and extended to incorporate invariances (Section D.1) and additive structure (Section D.2).

### D.1    Incorporating (permutation) invariance

In combinatorial BO, it is not uncommon for the objective function $f$ to possess certain symmetries, with permutation invariance perhaps being the most prevalent. When these symmetries are known, incorporating them into the kernel can significantly improve predictive accuracy (Borovitskiy et al., 2023), and therefore also sample efficiency.

Formally, let $G$ be a group acting on $\mathcal{X}$, then a function $f : \mathcal{X} \to \mathbb{R}$ is called $G$-invariant if

$$f(g(\mathbf{x})) = f(\mathbf{x}) \quad \forall g \in G \quad \forall \mathbf{x} \in \mathcal{X}.$$

Essentially, this means that $f$ remains unchanged under the action of any element of $G$ on $\mathcal{X}$. Now, for $f \sim \mathrm{GP}(0, k)$, we have that $f$ is $G$-invariant if and only if $k$ is $G$-invariant, which we define as being

$$k(\mathbf{x}, \mathbf{x}') = k(g_1(\mathbf{x}), g_2(\mathbf{x}')) \quad \forall g_1, g_2 \in G \quad \forall \mathbf{x}, \mathbf{x}' \in \mathcal{X}.$$

We refer the reader to Ginsbourger et al. (2013) for a proof.

Given a finite group $G$, Duvenaud (2014) propose three different methods to convert a kernel $k$ into its $G$-invariant version $k_{/G}$:

1.  summation

$$k_{\mathrm{sum}}(\mathbf{x}, \mathbf{x}') = \sum_{g \in G} \sum_{g' \in G} k(g(\mathbf{x}), g'(\mathbf{x}')),$$

2.  projection onto a fundamental domain

$$k_{\mathrm{proj}}(\mathbf{x}, \mathbf{x}') = k(A_G(\mathbf{x}), A_G(\mathbf{x}')),$$

3.  multiplication:

$$k_{\mathrm{prod}}(\mathbf{x}, \mathbf{x}') = \prod_{g \in G} \prod_{g' \in G} k(g(\mathbf{x}), g'(\mathbf{x}')).$$

Here, $A_G$ is a mapping used to transform $\mathbf{x}$ into a canonical representation $A_G(\mathbf{x})$ under $G$. For example, for the group $\mathrm{S}_n$ of permutations of a size $n$ set, Duvenaud (2014) suggest the fundamental domain $\{x_1, \ldots, x_n : x_1 \leq \cdots \leq x_n\}$, which can easily be mapped to by sorting $\mathbf{x}$ in ascending order. For permutation invariance, this sorting method is often much faster to execute than summation or multiplication, making it the preferred method of choice.

**Padding for permutation invariance**    Although sorting $\mathbf{x}$ will indeed result in a permutation-invariant kernel, it does not necessarily lead to good distances between different points. For example, consider the following two vectors, before and after sorting:

$$\mathbf{x} = [0, 0, 0, 1, 1, 2, 3, 3, 4, 4] \to [0, 0, 0, 1, 1, 2, 3, 3, 4, 4]$$
$$\mathbf{x}' = [4, 4, 0, 1, 1, 2, 3, 3, 4, 4] \to [0, 1, 1, 2, 3, 3, 4, 4, 4, 4].$$

Here, we expect both vectors to have a relatively low Hamming distance, since only the first two variables have been changed between $\mathbf{x}$ and $\mathbf{x}'$. However, after sorting, both vectors look very different, and their Hamming distance is quite large (i.e. 7).

To overcome this problem, we propose a simple trick: *padded sorting*. To do so, we introduce the new category "∗", and add it to our existing set of categories. In our previous example, all vectors in $\{0, \ldots, 4\}^{10}$ will then be mapped into $\{*, 0, \ldots, 4\}^{5*10}$ like this:

$$\mathbf{x} = [0, 0, 0, 1, 1, 2, 3, 3, 4, 4] \rightarrow [0, 0, 0, *, *, *, *, *, *, *, 1, 1, *, \cdots, *, 4, 4, *, *, *, *, *, *, *, *]$$

$$\mathbf{x}' = [4, 4, 0, 1, 1, 2, 3, 3, 4, 4] \rightarrow [0, *, *, *, *, *, *, *, *, *, 1, 1, *, \cdots, *, 4, 4, 4, 4, *, *, *, *, *, *].$$

As a result, the Hamming distance (after padded sorting) is now $2 + 0 + 0 + 0 + 2 = 4$, which is much more reasonable. In fact, with padded sorting, the distance between two sequences is something very intuitive: for every category in $\{1, \ldots, 4\}$, we compute the difference between the number of entries of this category in the first vector and the number of entries of this category in the second vector, and then we sum these numbers up. Although this trick is relatively simple, it has never been introduced in the Bayesian optimization literature (to the best of our knowledge).

In Appendix E.5, we use synthetic permutation-invariant functions to demonstrate the power of permutation-invariant heat kernels, as well as the increase in performance by switching to padded sorting. In Appendix F, we showcase the potential of permutation-invariant heat kernels for solving real-world problems, with an example on neural architecture search.

## D.2 Incorporating additive structure

Incorporating additive structure has proven successful for tackling high-dimensional problems or obtaining explainable models (Duvenaud et al., 2011; Durrande et al., 2012; Lu et al., 2022), and may thus prove useful in a combinatorial context. In this section, using our derivation from Appendix C, we extend the heat-kernel framework to such additive-based structures (in a straightforward way).

In Appendix C, we started from the most general combinatorial kernel and applied two simple assumptions (Assumptions 15 and 16), naturally obtaining heat kernels expressed as

$$k(\mathbf{x}, \mathbf{x}') = \prod_{i=1}^{n} k_i (x_i, x_i'), \tag{9}$$

with

$$k_i (x_i, x_i') = \begin{cases} v_i & \text{if } x_i = x_i' \\ c_i & \text{if } x_i \neq x_i' \end{cases}, \quad \frac{c_i}{v_i} \in \left( \frac{-1}{g_i - 1}, 1 \right) . \tag{10}$$

Now, we relax Assumption 15: we replace the product decomposition of Equation 9 by a more general decomposition

$$k(\mathbf{x}, \mathbf{x}') = f \left( k_1 (x_1, x_1'), \ldots, k_n (x_n, x_n') \right).$$

Here, some care needs to be exerted in picking a decomposition $f$, such that the obtained kernel $k(\mathbf{x}, \mathbf{x}')$ remains valid (i.e. symmetric positive semi-definite). Using this general decomposition $f$, we obtain an expressive class of kernels, which spans many well-known combinatorial kernels (including all Hamming or $\Phi$-kernels from Section 3.4).

**Connection to `CoCaBO` and `RDUCB`**  To extend the heat-kernel framework to additive structures, we keep the base kernels $k_i (x_i, x_i')$ from Equation 10 and select $f$ to be an additive function. For example, if we choose $f$ to be a simple first-order sum, we obtain

$$k(\mathbf{x}, \mathbf{x}') = \sum_{i=1}^{n} k_i (x_i, x_i'),$$

which is exactly the (additive) kernel used in `CoCaBO` (Ru et al., 2020a). Similarly, to obtain `RDUCB`'s random-decomposition kernel, we select $f$ to be

$$k(\mathbf{x}, \mathbf{x}') = \sum_{c \in C} \prod_{i \in c} k_i (x_i, x_i'),$$

with $c \subseteq \{1, \ldots, n\}$ and $C$ a (random) collection of different components $c$. Here, if two dimensions $i$ and $j$ do not appear together in any of the sets $c$, the kernel will not model interactions between them (Ziomek and Ammar, 2023). By choosing other (additive) functions for $f$, we have a principled way of coming up with new combinatorial kernels. We give an example in the following paragraph.

**New explainable kernel**   In the continuous Gaussian-process literature, there exist a number of additive-based kernels with an "explainable" or "interpretable" character. With the above framework, it becomes easy to extend them to combinatorial input spaces: keep the combinatorial base-kernels from Equation 10, and select the same additive decomposition $f$ as the original (continuous) kernel. For example, to adapt the explainable (additive) kernel from Duvenaud et al. (2011), we choose

$$k(\mathbf{x}, \mathbf{x}') = \sum_{d=1}^{n} \sigma_d \sum_{|c|=d} \prod_{i \in c} k_i \left( x_i, x_i' \right),$$

with $c \subseteq \{1, \ldots, n\}$. Here, the $\sigma_d$ are hyperparameters that will indicate how strong the effect is of different degrees of interactions, thus giving the kernel an "explainable" character. Our principled approach is general and can therefore be used to extend many other (continuous) explainable kernels to combinatorial input spaces. One promising candidate is the Orthogonal Additive Kernel (`OAK`) from Lu et al. (2022), which presents an improved version of the above kernel from Duvenaud et al. (2011). Although an interesting direction for further research, experimenting with these new kernels falls outside the scope of this paper.

# E   `MCBO` experiments

## E.1   Overview of benchmark problems

**SFU functions (20D)**   The `MCBO` benchmark (Dreczkowski et al., 2024) contains 21 test functions from Simon Fraser University (`SFU`), which have been generalized to $d$-dimensional domains (Surjanovic and Bingham, 2013). Moreover, in `MCBO`, these continuous functions have been discretized by only allowing the model to evaluate points on a regular grid (with $n$ values per dimension). These functions cover a range of optimization difficulties such as steep ridges, many local minima, valley-shape functions, and bowl-shaped functions. Out of these 21 `SFU` test functions, we select 5 permutation-invariant functions with different properties, and choose $d = 20$ and $n = 11$ as in Dreczkowski et al. (2024). Therefore, the benchmarks feature 20 categorical variables with 11 values each.

**Pest Control (25D)**   The `Pest Control` benchmark features 25 categorical variables with 5 values each, representing a chain of pest control stations (Oh et al., 2019). It models a dynamic optimization problem where pesticide prices decrease with frequent purchases, while effectiveness diminishes with repeated use due to pest tolerance. The objective is to minimize both pesticide costs and crop damage across the control chain. `Pest Control` has been included as benchmark in `COMBO` (Oh et al., 2019), `CASMOPOLITAN` (Wan et al., 2021), `BODi` (Deshwal et al., 2023), `Bounce` (Papenmeier et al., 2023) and `MCBO` (Dreczkowski et al., 2024).

**Contamination Control (25D)**   The `Contamination Control` benchmark features 25 binary variables, representing whether to quarantine uncontaminated food products (Hu et al., 2010). Similar to `Pest Control`, it features sequential decision-making with dynamic interactions, but represents a slightly simpler problem due to the use of binary rather than categorical variables. The objective function balances minimizing both the number of contaminated products reaching consumers and the costs associated with preventive quarantine measures. `Contamination Control` has been included as benchmark in `COMBO` (Oh et al., 2019), `CASMOPOLITAN` (Wan et al., 2021) and `Bounce` (Papenmeier et al., 2023).

**LABS (50D)**   The Low-Autocorrelation Binary Sequences (`LABS`) benchmark features 50 binary variables, representing a sequence $S$ of length 50 with elements in $\{-1, 1\}$. The objective is to find a sequence that minimizes the energy function $E(S) = \sum_{k=1}^{n-1} C_k^2(S)$, where $C_k(S)$ represents the autocorrelation at shift $k$. Alternatively, the objective can be expressed as maximizing the merit factor $F(S) = \frac{n^2}{2E(S)}$. This benchmark has practical applications in diverse fields, including communication engineering, radar and sonar ranging, satellite applications, and digital signal processing (Packebusch and Mertens, 2016). `LABS` has been included as benchmark in `BODi` (Deshwal et al., 2023) and `Bounce` (Papenmeier et al., 2023).

**MaxSAT (60D)**    The `MaxSAT` benchmark features 60 binary variables, representing the booleans in a weighted Maximum SATisfiability (`MaxSAT`) problem (Bacchus et al., 2018). The objective is to assign the boolean values such that the sum of the weighted clauses is maximized. Following `Bounce` (Papenmeier et al., 2023), we normalize these weights to have zero mean and unit standard deviation. `MaxSAT` has been included as benchmark in `COMBO` (Oh et al., 2019), `CASMOPOLITAN` (Wan et al., 2021), `BODi` (Deshwal et al., 2023) and `Bounce` (Papenmeier et al., 2023).

**Cluster Expansion (125D)**    The `Cluster Expansion` benchmark features 125 binary variables, and is derived from a real-world MaxSAT instance in materials science (Bacchus et al., 2018). As with the `MaxSAT` benchmark, the objective is to maximize the total weight of satisfied clauses, while treating the problem as a black box. To maintain consistency with `MaxSAT`, we also normalize the weights for `Cluster Expansion`, whereas this is not done in `Bounce` (Papenmeier et al., 2023). To the best of our knowledge, `Cluster Expansion` was introduced as a new combinatorial BO benchmark in `Bounce` (Papenmeier et al., 2023).

### E.2    Implementation details

The `MCBO` benchmark, introduced in Dreczkowski et al. (2024), consists of a collection of problems and algorithms for combinatorial BO. The benchmark is open-source (with MIT license), and allows us to perform all the experiments of Section 4 and Appendix E with minimal changes. The original benchmark is available at https://github.com/huawei-noah/HEBO/tree/master/MCBO, and our modified version can be found in the supplementary material. To prevent bias or overfitting, our experimental set-up relies as much as possible on the default values present in `MCBO`. We highlight some key elements of the configuration below.

**Experimental set-up**    We use 20 initialization points, opt for a batch size of 1 and allow up to 200 iterations. As hyperparameter optimizer and acquisition function, we use the ubiquitous Adam optimizer (Kingma and Ba, 2015) and Expected Improvement (EI) (Močkus, 1975; Jones et al., 1998), respectively. To optimize the acquisition function in the pipelines of Figures 1, 4, 6 and 8, we use a genetic algorithm and include a trust region (see Appendices F and G of Dreczkowski et al. (2024) for a detailed description). No priors were imposed on the hyperparameters, and therefore we use MLE and not MAP inference. We standardize all black-box function values using the mean and standard deviation of all previously observed values, and transform it back at prediction. The experiments are repeated across 20 random seeds, except for `Contamination Control` and `LABS`, which are noisier and therefore require double the amount of runs (i.e. 40 random seeds). Similarly, the biological and logic-synthesis datasets from Figures 5 and 6 require triple the amount of runs (i.e. 60 random seeds). Moreover, due to its significant computational requirements, we have halved the number of seeds for pipelines involving the `SSK`, as well as stopped their runs at 100 iterations on `Cluster Expansion`. The `MCBO` plots in this paper display the mean and standard error of the mean (with respect to the seeds) as a solid/dotted line and shaded area, respectively.

Out of all our baselines in Section 4 and Appendix E, `Bounce` (Dreczkowski et al., 2024) is the only one which is not implemented in `MCBO`. Therefore, we instead rely on the authors' open-source implementation (with MIT license), available at https://github.com/LeoIV/bounce. Because `Bounce` first explores lower-dimensional subspaces, we adhere to the authors' recommended set-up of 5 initialization points. This is visible on Figures 2 and 5, where `Bounce` starts with lower objective-values than the other baselines (at around 20 iterations).

**Relocation of the optima**    Following the approach in `Bounce` (Papenmeier et al., 2023), we relocate the optima (indicated by [reloc.]) using a fixed randomization procedure for each benchmark for all algorithms and repetitions. For binary problems, we flip each input variable independently with probability 0.5. For categorical problems, we randomly permute the order of the categories.

**Computational resources**    We were able to run all evaluated baselines on single-core CPUs with 1GB of RAM. The only notable exception is `SSK` (or `BOSS`), which is prohibitively slow on CPUs and was run on NVIDIA RTX6000 GPUs with 24GB of RAM. Table 1 gives an overview of the amount of time needed to run each method.

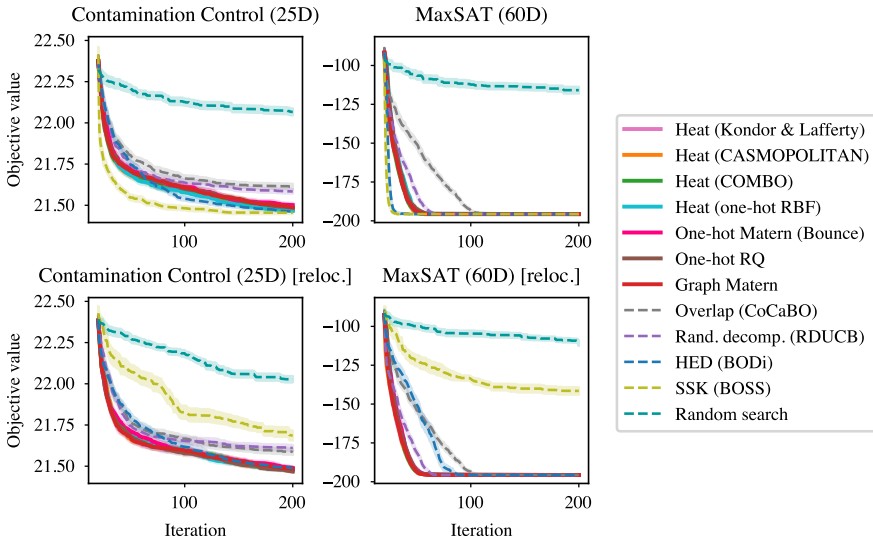

Figure 4: Extension of Figure 1 (when keeping the pipeline fixed and varying only the kernel choice, our unifying framework becomes visible empirically).

### E.3 Additional synthetic tasks

Figures 4 and 5 extend the results of Figures 1 and 2, respectively, to two new problems: `Contamination Control` and `MaxSAT`. These extended results fit the three takeaways presented in Section 4:

1. all heat (and even Hamming) kernels obtain nearly indistinguishable performance,
2. `BODi` (`HED`) and `BOSS` (`SSK`) experience a significant drop in performance after relocation of the optima,
3. a simple and fast pipeline, relying on heat kernels, achieves state-of-the-art performance.

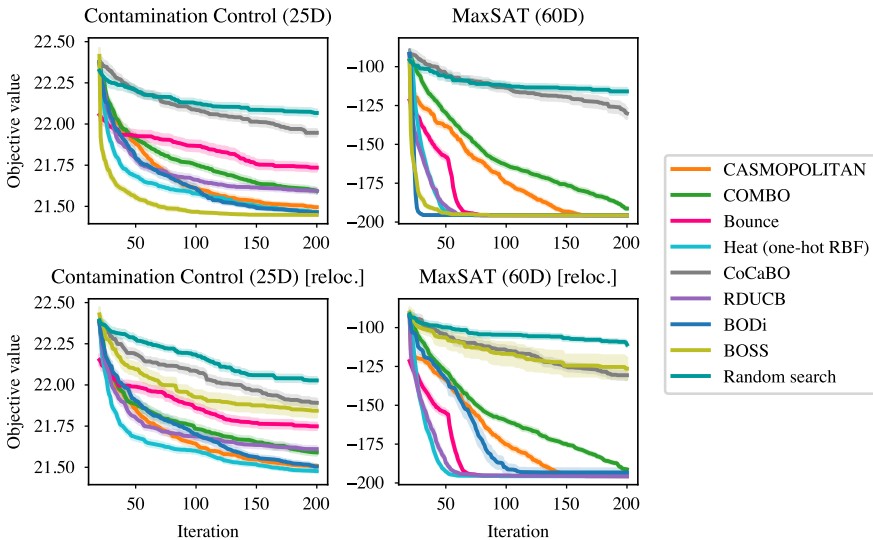

Figure 5: Extension of Figure 2 (a fast and simple pipeline, relying on heat kernels, achieves state-of-the-art results).

### E.4 Biological and logic-synthesis tasks

Figures 6 and 7 extend the results of Figures 1 and 2, respectively, to four new problems: `Antibody Design`, `AIG/MIG Optimization`, and `RNA Inverse Folding`. Here, the only exception is RDUCB, which was not evaluated on `Antibody Design` because its implementation (by the original authors) does not support input constraints (which are needed for this task). We refer to Dreczkowski et al. (2024) for a detailed description of all four problems.

These extended results on four new datasets mostly fit the three takeaways presented in Section 4, as well as lead to new insights:

1. On all four datasets, all Hamming kernels—indicated by solid lines in Figure 6—obtain nearly indistinguishable performance (as is the case for the other datasets presented in the paper).

2. On all four datasets, `SSK (BOSS)` and `HED (BODi)` do not seem particularly sensitive to relocation of the optima (a similar behavior was observed on `LABS` in Figures 1 and 2).

3. On all four datasets, our fast and simple heat-kernel pipeline achieves identical performance to `Bounce`, which is currently the state-of-the-art method. Interestingly, although both methods (`Bounce` and ours) are among the top performers on `AIG/MIG Optimization`, this is not the case on `Antibody Design` and `RNA Inverse Folding`.

   On these two datasets, there is a new trend: additive kernels, namely `Overlap (CoCaBO)` and `Rand. decomp. (RDUCB)`, are able to outperform Hamming kernels for the first time. This is not necessarily surprising: in related biological tasks, there exists evidence that first- and second-order interactions lead to strong predictive performance (Domingo et al., 2018; Faure et al., 2024). The `MCBO` paper (Dreczkowski et al., 2024) also contains potential evidence for this low-order hypothesis: a genetic algorithm, which only mutates a few variables at a time and does not inherently model high-order interactions, performs remarkably well on `RNA Inverse Folding`, significantly outperforming all BO methods.

   If such additive structure is known a priori, one can easily incorporate it into the heat-kernel, leading to methods such as (but not limited to) `RDUCB` and `CoCaBO` (see Appendix D.2). Nevertheless, besides the kernel, the acquisition-function optimizer seems to play an important role for these two biological datasets: `CASMOPOLITAN` (simple heat kernel) outperforms `CoCaBO` (additive-based heat kernel), although their respective kernels with a fixed acquisition-function optimizer lead to do the opposite ranking.

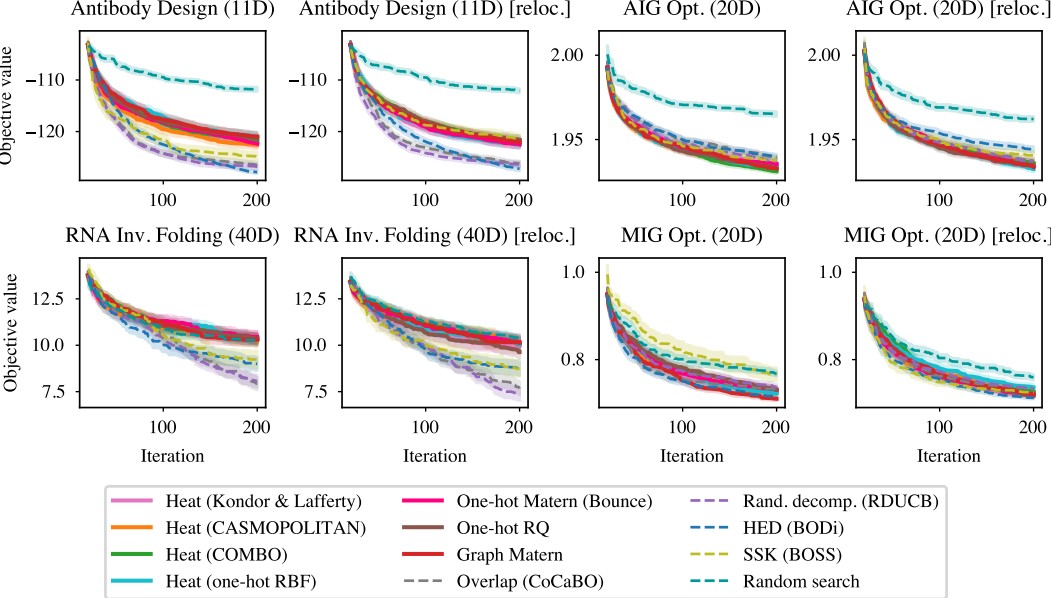

Figure 6: Extension of Figure 1 (when keeping the pipeline fixed and varying only the kernel choice, our unifying framework becomes visible empirically).

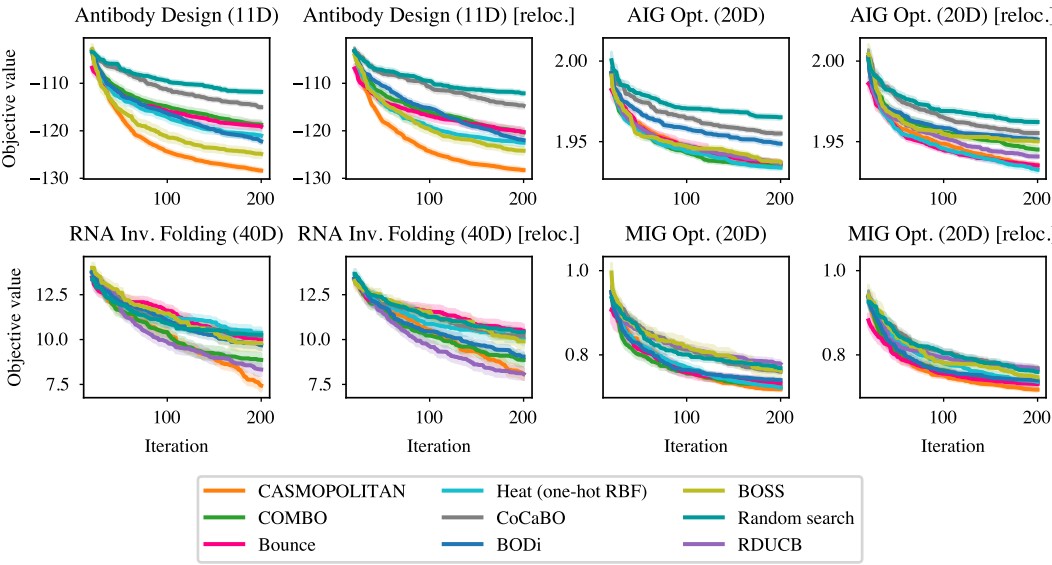

Figure 7: Extension of Figure 2 (a fast and simple pipeline, relying on heat kernels, matches the state-of-the-art method `Bounce`).

### E.5 Permutation-invariant `SFU` functions

In Section 4.2, we discuss the `SSK`'s (or `BOSS`') sensitivity to the location of the optima and present a simple hypothesis: the `SSK` favors points with repeated categorical values. Although a complete theoretical analysis of the mechanisms behind the `SSK`'s bias lies outside the scope of this paper, we do present an intuitive example for our hypothesis. Consider a substring $\mathbf{s}_1 = [1, 1]$ compared to the string $\mathbf{x}_1 = [1, 1, 1, 1]$, and another substring $\mathbf{s}_2 = [1, 2]$ compared to the string $\mathbf{x}_2 = [2, 1, 2, 1]$. Here, both substrings contain similar categorical values as their corresponding strings, but the `SSK` will generate six matches between $\mathbf{s}_1$ and $\mathbf{x}_1$, and only one match between $\mathbf{s}_2$ and $\mathbf{x}_2$. Therefore, we hypothesize that the `SSK` will have a biased perception of distance when it comes to points with repeated categorical values, due to these points accumulating a disproportionate amount of matches.

To empirically analyze this theoretical hypothesis, we rely on synthetic permutation-invariant functions, for which the global optima consist of repeated categorical values. As expected, Figure 8 displays a clear degradation of performance for `SSK` after relocation of the optima. This result is consistent with our hypothesis, since the random relocation means the optima will not necessarily consist of repeated categorical values.

Additionally, if we know the problem is permutation-invariant, we can instead rely on the techniques described in Appendix D.1. In Figure 8, `Heat (proj.)` refers to $k_{\text{proj}}(\mathbf{x}, \mathbf{x}')$ from Duvenaud (2014), which in this case amounts only to sorting the data before uing a heat kernel. As such, `Heat (pad. proj.)` represents our "padded sorting" approach proposed in Appendix D.1. From Figure 8, we conclude that padded sorting is a significant improvement over regular sorting. Moreover, even when the objective functions are permutation-invariant, there is no need to use the computationally intensive `SSK`, since `Heat (pad. proj.)` achieves similar performance on all five functions (except maybe for `Ackley Function`).

**Including "strategic" features**   When the objective function is invariant to groups other than the permutation group, the general techniques from Appendix D.1 can be used, whereas the `SSK` will not work. However, depending on the invariance group, these techniques can sometimes be computationally expensive. A more lightweight alternative, though perhaps less principled, is to include additional "strategic" features. For example, in the `Bounce` paper (Papenmeier et al., 2023), the authors argue that `BODi` achieves its performance mostly due to its way of doing "uniform" sampling of anchor points, which turns out to be heavily skewed towards points with a certain type of structure. As a consequence, `BODi`'s `HED` embeddings or features contain a clear distance metric to anchor points with a certain structure, which can become a useful signal to navigate towards certain

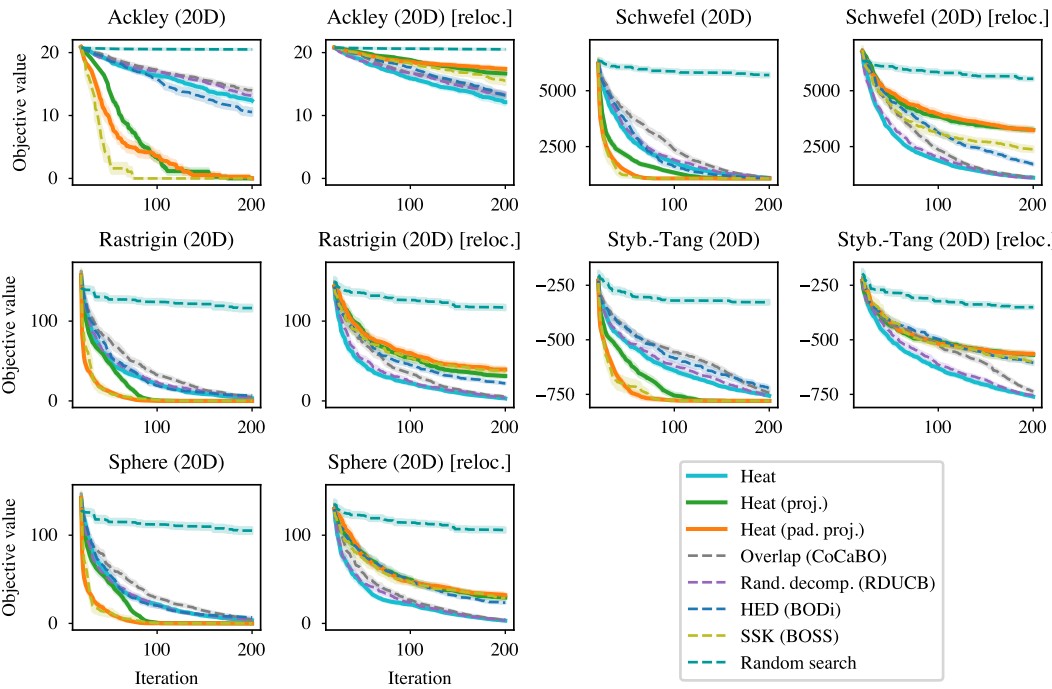

Figure 8: Our padded version of $k_{\text{proj}}(\mathbf{x}, \mathbf{x}')$, proposed in Appendix D.1, outperforms the non-padded version from Duvenaud (2014), as well as matches the `SSK` on nearly all problems.

optima. The same idea can easily be integrated within our heat-kernel framework: sample anchor points according to the desired type of structure, and add the relevant features (i.e. the Hamming distance to these anchor points) to the existing heat-kernel. We believe this general idea might be an interesting direction for further research, but leave it to future works.

### E.6 Wall-clock time

Table 1 displays the wall-clock time (in seconds) of different pipelines. This speed measurement is concerned only with the algorithms themselves, and therefore does not include the time needed for evaluation of the objective function. All baselines are measured on a single-core CPU, except for `BOSS`, whose speed measurements should be interpreted with caution given that they are based on highly-vectorized code run on a GPU. A clear pattern is visible for nearly all methods across the different benchmarks: as the dimensionality of the problem increases, so does the wall-clock time. Suprisingly, this is not the case for `Bounce`, where the wall-clock time stays roughly constant across varying dimensionalities. We hypothesize that this is due to `Bounce`'s nested embeddings, which significantly reduce the dimensionality of the problem for most of the iterations.

Figure 3 summarizes the results from Table 1 as follows: divide the wall-clock time of each method by that of the fastest method (i.e. `CoCaBO`), and then take the average of these ratios across all benchmarks. Using this summary, we can clearly observe the benefits of our faster implementation from Section 3.3.3: `one-hot` RBF is more than $2\times$ faster than `COMBO`'s kernel (on average across all five problems). Additionally, the simple heat-kernel pipeline proposed in Section 4.3 is also fast, with only minimal overhead compared to `CoCaBO` or `Bounce`.

## F `NASLib` experiments

To highlight the power of group-invariant heat kernels, we provide a simple example on a real-world permutation-invariant problem: Neural Architecture Search (NAS). To the best of our knowledge, in the subfield of Bayesian optimization for NAS, the current state-of-the-art method is Ru et al. (2020b), which uses a Gaussian process with the Weisfeiler–Lehman (`WL`) kernel from Shervashidze et al. (2011). In this appendix, we benchmark a (permutation-invariant version of) heat kernels against the

Table 1: The wall-clock time (in seconds) increases together with the dimensionality of the problem.

| Method | Pest Control (25D) | Contam. Control (25D) | LABS (50D) | MaxSAT (60D) | Cluster Exp. (125D) |
|---|---|---|---|---|---|
| CoCaBO | 227.40 ± 8.32 | 233.18 ± 5.61 | 369.27 ± 6.56 | 435.30 ± 8.88 | 857.51 ± 15.44 |
| Bounce | 548.42 ± 13.57 | 448.51 ± 6.56 | 496.14 ± 12.97 | 434.04 ± 11.57 | 471.77 ± 9.25 |
| Heat (one-hot RBF) | 403.97 ± 9.42 | 375.80 ± 6.54 | 565.01 ± 5.92 | 610.68 ± 9.20 | 1082.23 ± 15.71 |
| RDUCB | 413.60 ± 3.04 | 394.50 ± 4.15 | 659.97 ± 8.96 | 812.95 ± 7.47 | 1593.04 ± 15.98 |
| CASMOPOLITAN | 534.19 ± 6.16 | 521.41 ± 10.50 | 727.54 ± 13.60 | 793.97 ± 21.80 | 1298.76 ± 79.95 |
| BODi | 615.64 ± 9.53 | 577.63 ± 12.40 | 733.76 ± 7.12 | 791.20 ± 12.76 | 1131.67 ± 24.87 |
| Heat (COMBO) | 764.70 ± 9.06 | 691.53 ± 11.67 | 1173.37 ± 18.92 | 1367.53 ± 13.52 | 2672.81 ± 25.69 |
| BOSS | 1782.29 ± 13.33 | 4662.90 ± 1648.68 | 5376.22 ± 57.55 | 7370.48 ± 98.19 | / |
| COMBO | 6163.21 ± 144.18 | 4522.70 ± 66.74 | 11982.25 ± 195.49 | 19359.63 ± 766.23 | 49606.75 ± 1630.83 |

`WL` graph kernel, following the set-up of Ru et al. (2020b) using the `NASLib` package (Mehta et al., 2022).

## F.1 Overview of benchmark problems

**NAS-Bench-101**   Ying et al. (2019) introduce `NAS-Bench-101`, a tabular dataset containing 423,624 unique convolutional neural network architectures exhaustively generated from a fixed graph-based search space and evaluated on CIFAR-10. Each architecture is represented as a directed acyclic graph with up to 9 vertices and 7 edges, where operations include 3×3 convolution, 1×1 convolution, and 3×3 max-pooling. The dataset provides over 5 million trained models, with each architecture trained multiple times at various training budgets (4, 12, 36, and 108 epochs) and evaluated three times each, reporting metrics including training/validation/test accuracy, number of parameters, and training time.

**NAS-Bench-201**   Dong and Yang (2020) introduce `NAS-Bench-201`, which extends the scope of reproducible NAS research by providing a cell-based search space with 15,625 unique architectures evaluated across three datasets: CIFAR-10, CIFAR-100, and ImageNet-16-120. Each architecture is represented as a densely-connected directed acyclic graph with 4 nodes, where edges are selected from 5 operations: none, skip connection, 1×1 convolution, 3×3 convolution, and 3×3 average pooling. Unlike NAS-Bench-101, this benchmark is applicable to almost any up-to-date NAS algorithm, including parameter sharing and differentiable methods, and provides fine-grained diagnostic information such as epoch-by-epoch training/validation/test loss and accuracy.

## F.2 Implementation details

The `NASLib` benchmark, introduced in Mehta et al. (2022), consists of a collection of problems and algorithms for neural architecture search. The benchmark is open-source (with Apache 2.0 license), and allows us to perform all the experiments of Appendix F with minimal changes. The original benchmark is available at https://github.com/automl/NASLib, and our modified version can be found in the supplementary material. To prevent bias or overfitting, our experimental set-up relies as much as possible on the default values present in `NASLib` and Ru et al. (2020b). We highlight some key elements of the configuration below.

**Experimental set-up**   We use 10 initialization points, opt for a batch size of 5 and allow up to 150 iterations. As hyperparameter optimizer and acquisition function, we use the ubiquitous Adam optimizer (Kingma and Ba, 2015) and Expected Improvement (EI) (Močkus, 1975; Jones et al., 1998), respectively. We transform all black-box function values (i.e. the test errors) by converting them to log-scale and standardizing them using the mean and standard deviation of all previously observed values. As above, we transform these values back at prediction. All experiments are repeated across 20 random seeds, and all `NASLib` plots in this paper display the median and standard errors (with respect to the seeds) as a solid line and shaded area, respectively. To optimize the acquisition function, we use a pool size of 200, where one half is generated from random sampling and the other half is generated from mutating the top-10 best performing architectures already queried (see Appendix F.2

of Ru et al. (2020b) for a detailed description). No priors were imposed on the hyperparameters, and therefore we use MLE and not MAP inference.

**Permutation-invariance for graphs**  In order to apply the heat kernel to the labeled graphs of varying nodes found in `NAS-Bench-101/201`, we transform the input space following Borovitskiy et al. (2023). More specifically, to deal with the varying sizes, we pad all graphs with additional disconnected nodes, such that all graphs now have the same order as the graph in the dataset with the maximal number of nodes. Additionally, to deal with the labels of the graph, we assign certain types of nodes to pre-specified groups of vertices, thus aligning different data samples better (see Figure 5 of Borovitskiy et al., 2023). Lastly, to account for graph automorphisms, we use the group-invariant kernel $k_{\mathrm{sum}}(\mathbf{x}, \mathbf{x}')$ from Section D.1 and approximate the large sum as in Borovitskiy et al. (2023), namely using Monte Carlo sampling:

$$k_{\mathrm{sum}}(\mathbf{x}, \mathbf{x}') \approx \frac{1}{|S|^2} \sum_{\sigma_1 \in S} \sum_{\sigma_2 \in S} k\left(\sigma_1(\mathbf{x}), \sigma_2(\mathbf{x}')\right), \qquad S \subseteq \mathbf{S}_n, \qquad S \ni \sigma \overset{\mathrm{i.i.d.}}{\sim} \mathrm{U}(\mathbf{S}_n),$$

where $\mathbf{S}_n$ denotes the set of all permutations and $\mathrm{U}(\mathbf{S}_n)$ is the uniform distribution over $\mathbf{S}_n$. For all `NASLib` experiments in this paper, we have used $|S| = 200$.

### F.3    Regression problems and Bayesian optimization

As done in Ru et al. (2020b), we analyze the regression and Bayesian-optimization performance of different kernels on `NASLib`. For the regression problems, we use different training sizes, ranging from 25 to 200 datapoints, and a test size of 200 datapoints. Here, all datapoints are sampled uniformly at random. Again, similarly to Ru et al. (2020b), we choose the Spearman's rank correlation between predicted and true accuracy as the performance metric, as what matters for selecting architectures in BO loops is their relative, and not absolute, performance. For Bayesian optimization, we provide a more detailed explanation of our set-up in Appendix F.2.

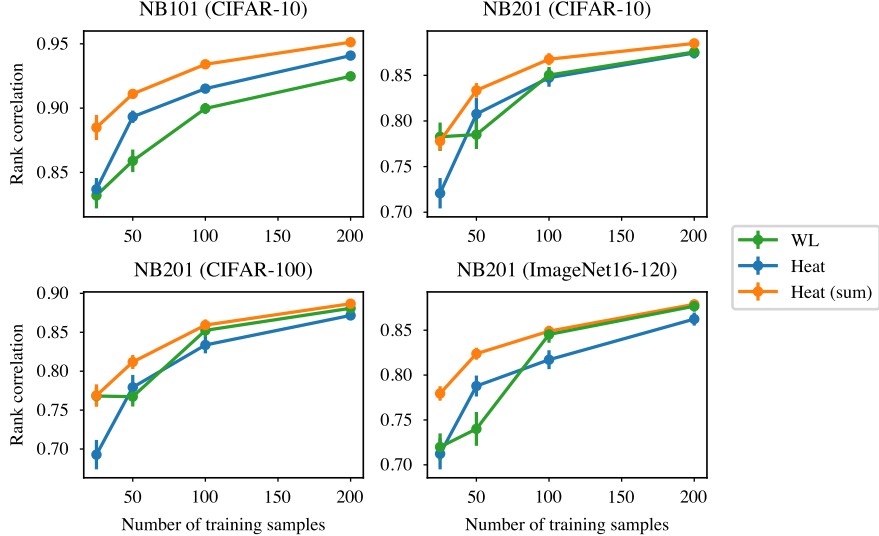

Figure 9: For regression problems on the `NASLib` benchmark, the `Heat` kernel performs comparably to the state-of-the-art `WL` kernel. An averaged version of this `Heat` kernel, indicated by `Heat (sum)`, outperforms the `WL` kernel.

In Figures 9 and 10, we compare three kernels: the state-of-the-art Weisfeiler–Lehman (`WL`) kernel, the "standard" heat kernel (`Heat`), and its permutation-invariant version $k_{\mathrm{sum}}(\mathbf{x}, \mathbf{x}')$, indicated by `Heat (sum)`. For this last one, we use the Monte Carlo approximation from Appendix F.2 with 200 random permutations. On all datasets, `Heat (sum)` outperforms `Heat`, and achieves similar or better performance compared to `WL`. We argue that this is a strong result, given that the `WL` kernel is designed specifically for graphs, whereas heat kernels are general and lead to strong performance on a range of different combinatorial structures.

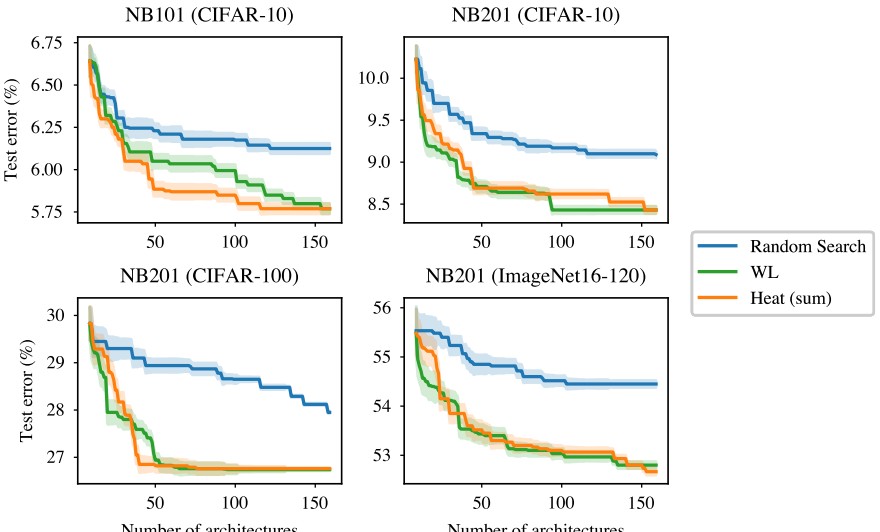

Figure 10: For Bayesian optimization on the `NASLib` benchmark, an averaged version of the `Heat` kernel, indicated by `Heat (sum)`, achieves similar performance to the state-of-the-art `WL` kernel.

