# OpenReview forum: "Omnipresent Yet Overlooked: Heat Kernels in Combinatorial Bayesian Optimization"
_NeurIPS.cc/2025/Conference — NeurIPS 2025 poster_

### Official Review · Reviewer_oZnX · 2025-06-30

**Clarity:** 3
**Significance:** 3
**Originality:** 3
**Rating:** 5
**Confidence:** 4

**Summary:**

The paper mainly presents an analysis to unify different kernels used commonly in categorical BO. The paper states that these common kernels, e.g. kernels in CASMOPOLITAN, COMBO, Bounce, CoCaBO, etc., are either heat kernels (diffusion kernels), or parts of a certain generalized class containing heat kernels. Notable claims include: (1) CASMOPOLITAN and COMBO are both fundamentally heat kernels, (2) CASMOPOLITAN and Bounce come from the class of Hamming-based heat kernel (3) COMBO comes from the class of graph-based heat kernel, (4) Hamming-based and graph-based kernels are equivalent under a non-trivial assumption, (5) heat kernels can be extended to relate to CoCaBO and RDUCB kernels. All claims are supported with theoretical analysis. Experiments demonstrate the equivalence of certain kernels, and show that a simple heat kernel pipeline can have equivalent performance to the state-of-the-art baselines.

**Questions:**

-	Given the simplicity of heat kernel pipeline, can the authors implement the discussed new ideas in Hamming-based approach in Sec. 3.4.1(via rational quadratic kernel), and similarly for Graph-based kernels in Sec. 3.4.2 (via graph Matern kernel and sum-of-inverse polynomial kernel)?
-	Is there any way to make heat kernel enjoy the same benefit of BODi and BOSS kernels such that the performance is also great in problems with special structure? It would be much better if a kernel can work well in both scenarios.

**Ethical Concerns:**

["NO or VERY MINOR ethics concerns only"]

**Final Justification:**

The authors answered adequately to all of my questions, therefore, I keep my original score for this paper, which is Accept.

**Limitations:**

Yes

**Paper Formatting Concerns:**

The format is good to me.

**Quality:**

4

**Strengths And Weaknesses:**

Strengths:

-	The paper proposes a unifying framework for many common kernels in categorical BO, which is very interesting and important to understand this field of BO.
-	The paper is well written and easy to follow.
-	There are many theoretical results that clearly state the assumptions and drawbacks.
-	Experimental results provide interesting insights on the equivalence of many kernels.

Weaknesses:

-	I think the authors could leverage the ideas and propose more effective kernels that can empirically outperform the state-of-the-art baselines. See my questions.
-	I do not think no-ordinal variables is a standard in combinatorial BO papers (line 350). In my opinion, many recent papers either treat ordinal variables as categorical or avoid addressing them due to the additional complexity involved, especially when their methods already demonstrate strong performance on categorical inputs. Hence the ordinal variables have received less attention. In fact, many real-world tasks require the ordinal variables, notably in HPO (number of trees or max depth in random forests, number of layers and units in neural networks, etc.). Examples in other fields include [1-3], which have all been used in BO works [4-5].
-	The empirical performance is only on par with the state-of-the-art, slightly reducing the contribution of the work.

[1] Kohira, Takehisa, et al. "Proposal of benchmark problem based on real-world car structure design optimization." Proceedings of the Genetic and Evolutionary Computation Conference Companion. 2018.

[2] Dreifuerst, Ryan M., et al. "Optimizing coverage and capacity in cellular networks using machine learning." ICASSP 2021-2021 IEEE International Conference on Acoustics, Speech and Signal Processing (ICASSP). IEEE, 2021.

[3] Wang, Boqian, et al. "Harnessing a novel machine-learning-assisted evolutionary algorithm to co-optimize three characteristics of an electrospun oil sorbent." ACS Applied Materials & Interfaces 12.38 (2020): 42842-42849.

[4] Daulton, Samuel, et al. "Bayesian optimization over discrete and mixed spaces via probabilistic reparameterization." Advances in Neural Information Processing Systems 35 (2022): 12760-12774.

[5] Daulton, Samuel, et al. "Multi-objective bayesian optimization over high-dimensional search spaces." Uncertainty in Artificial Intelligence. PMLR, 2022.

---

> ### Author Rebuttal · Authors · 2025-07-31
>
> We are gratified to see the reviewer found the paper to be well-written, and our unifying framework to be "very interesting and important". We thank the reviewer for this thoughtful response, and reply to the mentioned points below.
>
> >I do not think no-ordinal variables is a standard in combinatorial BO papers (line 350). In my opinion, many recent papers either treat ordinal variables as categorical or avoid addressing them due to the additional complexity involved, especially when their methods already demonstrate strong performance on categorical inputs. Hence the ordinal variables have received less attention. In fact, many real-world tasks require the ordinal variables, notably in HPO (number of trees or max depth in random forests, number of layers and units in neural networks, etc.). Examples in other fields include [1-3], which have all been used in BO works [4-5].
>
> We believe a slight misunderstanding took place, and thank the reviewer for bringing it to our attention. Since different sources rely on different definitions for _ordinal variables_, we now explicitly specify ours: "Ordinal data is a categorical, statistical data type where the variables have natural, ordered categories and _the distances between the categories are not known_." This is the main definition on Wikipedia [6], which cites [7] as being the source for this definition. We have carefully examined all the variables in the suggested benchmarks [1-3], and conclude, to the best of our knowledge, that none of them are ordinal variables (according to our definition). As a general example, we consider the number of layers in neural networks. Here, although the variable is discrete, there is a clear distance between the different categories (namely the Euclidean distance). Therefore, according to our definition, this is not an ordinal variable. For lack of a better word, we will refer to these discrete variables (containing an inherent notion of distance) as "discrete-quantitative variables".
>
> In light of the above clarification, we believe our claim holds true: "our experiments do not contain ordinal variables, but we note that this is standard in recent combinatorial BO papers" (line 350). Following this insightful discussion with the reviewer, we have updated the camera-ready version of the paper: we have included a more explicit categorization of the different possible discrete variables, as well as an expanded discussion of how existing works fit into this categorization.
>
> As pointed out by the reviewer, combinatorial BO papers do often contain discrete-quantitative variables. For such variables, we believe the associated kernels are rather straightforward: use any (continuous) isotropic kernel, along with an appropriate distance function (usually Euclidean). As such, these kernels can simply be seen as continuous kernels, which is why we did not explicitly address them in our unifying framework. Additionally, although discrete-quantitative variables can borrow their kernels from continuous BO, we note that this is not true for their acquisition-function optimizers, which need to be discrete (e.g. genetic algorithm). We are grateful to the reviewer for highlighting this additional type of discrete variable, and have included an additional paragraph in the camera-ready version that discusses this relevant topic.
>
> Lastly, for ordinal variables (according to our definition of the term), we would like to mention our answer provided to Reviewer `b9FW`. In this answer, we explain how to model ordinal variables using path graphs $\mathcal{G}_i$, therefore connecting them to our existing heat-kernel framework.
>
> [6] [https://en.wikipedia.org/wiki/Ordinal_data](https://en.wikipedia.org/wiki/Ordinal_data)
>
> [7] Agresti, Alan. _Categorical Data Analysis._ John Wiley & Sons (2002).
>
> >Given the simplicity of heat kernel pipeline, can the authors implement the discussed new ideas in Hamming-based approach in Sec. 3.4.1(via rational quadratic kernel), and similarly for Graph-based kernels in Sec. 3.4.2 (via graph Matern kernel and sum-of-inverse polynomial kernel)?
>
> We have implemented two of the above three kernels: the Hamming-based rational-quadratic kernel and the graph-based Matérn kernel. To the best of our understanding, the sum-of-inverse polynomial kernel is fundamentally an ARD kernel, and turning this into a non-ARD kernel would not be sensible. Since our experimental evaluation considers only non-ARD kernels, we believe it would be confusing for the reader to add an ARD kernel, which is why we have omitted it.
>
> Unfortunately, the NeurIPS policy this year does not allow us to show or link images in any way whatsoever. Fortunately, in this case, it should not be a problem, as the results are easy to describe: on all seven benchmarks (including the additional two suggested by Reviewer `opDa`), the Hamming-based rational-quadratic kernel and the graph-based Matérn kernel achieve near-identical performance compared to the five other Hamming kernels evaluated in the paper. As a result, we are now left with one single rainbow-colored line, which includes seven different kernels that all obtain the exact same empirical performance. To the best of our knowledge, we believe we are the first to demonstrate the empirical (and partly theoretical) equivalence between these seven kernels for combinatorial BO, which were previously linked to distinct methods.
>
> Moreover, in Section 3.4 of the paper, we show that Hamming- and graph-based kernels can be seen as broader (generalized) versions of heat kernels, and that these two classes also coincide (for $|\mathcal{X}_1| = \ldots = |\mathcal{X}_n|$). With these new experiments, we now have additional empirical evidence for these theoretical results. We believe these experiments meaningfully improve the paper, and sincerely thank the reviewer for suggesting them.
>
> >Is there any way to make heat kernel enjoy the same benefit of BODi and BOSS kernels such that the performance is also great in problems with special structure? It would be much better if a kernel can work well in both scenarios.
>
> The short answer is "yes", and we see two ways of doing this: incorporating group-invariances, or including additional features.
>
> When it is known that the problem at hand has a certain type of structure, this structure can often be modeled by incorporating invariance to a certain group $G$ directly into the kernel. As explained in Section 3.5 and Appendix C.1, our heat-kernel framework allows for the incorporation of _any_ group-invariance, and this through a straightforward procedure. In a way, this is also what the `SSK` (kernel from `BOSS`) does, since it models certain string-invariances, which seems to be a powerful inductive bias for certain types of optima. In Figures 5 and 7, we empirically demonstrate the advantages of incorporating such group-invariances, resulting in state-of-the-art performance. However, in general, modeling invariances can quickly become expensive, as can be seen by `BOSS`' high computational cost in Figure 6 (and the fact that it needs highly-vectorized code and high-memory GPUs to get to this speed).
>
> A more lightweight alternative, though perhaps less principled, is to include additional "strategic" features. For example, in the `Bounce` paper [8], the authors argue that `BODi` achieves its performance mostly due to its way of doing "uniform" sampling of anchor points, which turns out to be heavily skewed towards points with a certain type of structure. As a consequence, `BODi`'s `HED` embeddings or features $\phi_{\mathbf{A}}(\mathbf{x})$ contain a clear distance metric to anchor points with a certain structure, which can become a useful signal to navigate towards certain optima. The same idea can easily be integrated within our heat-kernel framework: sample anchor points according to the desired type of structure, and add the relevant features (i.e. the Hamming distance to these anchor points) to the existing heat-kernel. We believe this general idea might be an interesting direction for further research, and have added it to the camera-ready version of the paper. Once again, we appreciate the reviewer's efforts to improve the paper and make it more complete.
>
> [8] Papenmeier, Leonard, Luigi Nardi, and Matthias Poloczek. _Bounce: Reliable High-Dimensional Bayesian Optimization for Combinatorial and Mixed Spaces_. NeurIPS (2023).

---

> > ### Comment · Reviewer_oZnX · 2025-08-05
> >
> > I thank the authors for their detailed response to my review. I think the response addresses my concerns, so I will keep my score for the paper as is.

---

### Official Review · Reviewer_ot94 · 2025-07-01

**Clarity:** 4
**Significance:** 4
**Originality:** 4
**Rating:** 6
**Confidence:** 4

**Summary:**

The paper presents a unifying review of methods in combinatorial BO. Authors show that the kernels of popular CASMOPOLITAN and COMBO methods are mathematically equivalent and that they both belong to a wider class of heat kernels. This finding has profound implications. First of all, it means that the only differences between CASMOPOLITAN and COMBO we empirically observed in the past were not coming from differences in surrogate models, which authors empirically demonstrate by running both methods with exact same settings and showing their performance is indistinguishable. Secondly, it means one can perform the computation of the COMBO kernel much faster, which authors demonstrate achieving 2x speedup. Lastly, authors show that a basic form of the heat kernel with the best performing acquisition optimiser and trust region is sufficient to achieve state-of-the-art performance across a number of problems.

**Questions:**

How do you think one could incorporate continuous variables into the heat kernel framework in the Mixed-space BO problem setting?

**Ethical Concerns:**

["NO or VERY MINOR ethics concerns only"]

**Final Justification:**

I did not have any major concerns regarding the submission, my questions were mostly points that I wanted to clarify. After authors clarified them, I maintain my positive evaluation of the work.

**Limitations:**

yes

**Quality:**

4

**Strengths And Weaknesses:**

Strengths:
- The paper unravels a vital connection between two well-known methods that the community was previously oblivious to.
- The paper supports this surprising theoretical result with convincing and comprehensive experiments
- The papers draws a number of important conclusions from their theoretical results that are incredibly useful to practitioners, for example, that given the same hyperparameters and acquisition optimiser, CASMOPOLITAN and COMBO will perform the same and that the computation of COMBO kernel can be massively sped up
- The paper provides clear guidance to practitioners, proposing a simple, yet powerful baseline
- The paper is accompanied by an extensive appendix, which discusses how to adopt the heat kernels to invariances or how to incorporate an additive structure into them

Weaknesses:
- I do not think there are any significant weaknesses, however, I believe the presentation of the paper could be slightly improved. A lot of important results and insights appear only in the appendix. Authors should think carefully about how to best use the additional page in camera-ready version and move the most important parts of the appendix there. It would also be nice to report the number of seeds used for each run in the main text.
- (typo): I believe equation 4 should have a minus inside the exponent

---

> ### Author Rebuttal · Authors · 2025-07-31
>
> We are excited and encouraged to see the reviewer's strong positive response, and are happy to reply to their questions below.
>
> >I do not think there are any significant weaknesses, however, I believe the presentation of the paper could be slightly improved. A lot of important results and insights appear only in the appendix. Authors should think carefully about how to best use the additional page in camera-ready version and move the most important parts of the appendix there. It would also be nice to report the number of seeds used for each run in the main text.
>
> We would like to thank the reviewer for highlighting this aspect. During this rebuttal period, we have taken great care to migrate the most important parts of the appendix to the additional page of the camera-ready version. We have also made sure to include the number of seeds of each run in the main paper.
>
> >(typo): I believe equation 4 should have a minus inside the exponent
>
> We politely disagree with the reviewer, and provide a short derivation below to explain our intuition. Here, we start with an RBF kernel applied to one-hot encoded data, namely $\mathbf{z}, \mathbf{z}' \in \\{0,1\\}$:
> $$\exp\left(-\frac{\lVert\mathbf{z} - \mathbf{z}'\rVert^2_2}{2\ell}\right).$$
> As expected, this kernel has a minus inside the exponent. Afterwards, we use the equivalence from Equation 1 in the paper:
> $$h(\mathbf{x}, \mathbf{x}') = \frac{\lVert\mathbf{z} - \mathbf{z}'\rVert_2^2}{2},$$
> where $h(\mathbf{x}, \mathbf{x}')$ is the Hamming distance between categorical variables $\mathbf{x}, \mathbf{x}' \in \mathcal{X}$. Using this equivalence, and explicitly writing out the Hamming distance in terms of Kronecker-delta functions $\delta(\cdot,\cdot)$, we get:
> $$
> \exp\left(-\frac{\lVert\mathbf{z} - \mathbf{z}'\rVert^2_2}{2\ell}\right) = \exp\left(-\frac{h\left(\mathbf{x}, \mathbf{x}'\right)}{\ell}\right)
> = \exp\left(-\frac{1}{\ell} \left[n - \sum_{i=1}^n \delta(x_i, x_i')\right] \right)
> \propto \exp\left(\sum_{i=1}^n \frac{\delta(x_i, x_i')}{\ell}\right),
> $$
> where the minus sign has now disappeared. If we replace $\ell$ by $\ell_i$ to allow for ARD, and re-parameterize with $\ell_i := n/\gamma_i$, we obtain
> $$\exp\left(\frac{1}{n}\sum_{i=1}^n \gamma_i \delta(x_i, x_i')\right),$$
> which is exactly the kernel proposed in Equation 1 of [1] (and presented in Equation 4 of our paper). We hope that this derivation was able to clarify the reason for the missing minus sign, and are happy to address any remaining questions the reviewer might have.
>
> [1] Wan, Xingchen, et al. _Think Global and Act Local: Bayesian Optimisation over High-Dimensional Categorical and Mixed Search Spaces._ ICML (2021).

---

> > ### Comment · Reviewer_ot94 · 2025-08-03
> >
> > I see, thank you for the explanation. I admit this was my mistake and equation 4 is correct. I remain very positive about the paper.

---

### Official Review · Reviewer_opDa · 2025-07-02

**Clarity:** 3
**Significance:** 3
**Originality:** 2
**Rating:** 5
**Confidence:** 5

**Summary:**

Recent works on combinatorial BO have studied the impact of acquision function optimisation, the use of trust regions and the surrogate models.
While GP-based approaches are SOTA in general combinatorial BO in the absense of prior information, there is still a gap in the understanding of the impact of the kernel choice
on the BO performance. In this work the authors aims at bridging this gap, not only by empirically assessing the merits of the standard combinatorial kernels
(proposed in COMBO, CASMOPOLITAN, ...) but by actually revealing that most of these standard kernels are somewhat equivalent and belong to a larger family of Hamming-based kernels.

**Questions:**

Please see above

**Ethical Concerns:**

["NO or VERY MINOR ethics concerns only"]

**Final Justification:**

Great paper and the rebuttal addressed all my comments.

**Limitations:**

ok

**Quality:**

3

**Strengths And Weaknesses:**

Strengths:
- The paper is clearly written and easy to follow.
- The unifying framework presenting Casmopolitan and COMBO kernels as heat kernels is theoretically sound, and offers a new perspective on the combinatorial kernel landscape.
- The experiments support the equivalence claim and the proposed pipeline is simple and fast.
- The proposed generalization to Hamming-based kernels open the way to new investigation.

Weakness:
- One could say the main results is simply derived given the results of Kondor and Lafferty on the diffusion kernel for the complete graph (though it is true that it was not exploited or noticed before)
- It could be interesting to consider more real-world black-boxes (such as the ones from the MCBO library); Please evaluate on those:


Minor comments:
- line 229: it should be "by choosing k", not "\mathbb{k}"
- Have a line style distinct for heat-based and non-heat-based to ease the analysis

Extra question:
- Is there a way to see the kernel from BODi as a heat kernel variation in some sense?

---

> ### Author Rebuttal · Authors · 2025-07-31
>
> We are pleased to see the reviewer found the paper to be "clearly written" and our framework to be "theoretically sound". We are grateful for the reviewer's input and will be addressing their comments.
>
> >It could be interesting to consider more real-world black-boxes (such as the ones from the MCBO library); Please evaluate on those:
>
> In the `MCBO` library, there a four categorical benchmarks left on which we did not evaluate: two biological benchmarks (`RNA Inverse Folding` and `Antibody Design`) and two logic-synthesis optimization benchmarks (`AIG Optimization` and `MIG Optimization`). Since `Antibody Design` and `MIG Optimization` contain much slower simulators (`MIG Optimization` can be up to $200.000\times$ slower), we focus our efforts in this limited time-window on `RNA Inverse Folding` and `AIG Optimization`. As a result, the camera-ready paper now also contains one biological and one logic-synthesis optimization task. We believe this makes the experimental section of the paper more complete, and are grateful to the reviewer for this suggestion.
>
> Unfortunately, the NeurIPS policy this year does not allow us to show or link images in any way whatsoever. Consequently, we use text to describe the main trends observed on the new plots:
> 1. On both `RNA Inverse Folding` and `AIG Optimization`, all Hamming kernels obtain nearly indistinguishable performance (as is the case for the other datasets presented in the paper).
> 2. On both datasets, `SSK` (`BOSS`) and `HED` (`BODi`) do not seem particularly sensitive to relocation of the optima (a similar behavior was observed on `LABS` in Figures 1 and 2 of the paper).
> 3. On both datasets, our fast and simple heat-kernel pipeline achieves identical performance to `Bounce`, which is currently the state-of-the-art method. Interestingly, although both methods (`Bounce` and ours) are the top performers on `AIG Optimization`, this is not the case on `RNA Inverse Folding`. On this last dataset, there is a new trend: additive kernels, namely `CoCaBO` and `RDUCB`, are able to outperform Hamming kernels for the first time. This is not necessarily surprising: in related biological tasks, there exists evidence that first- and second-order interactions lead to strong predictive performance [1-2]. The `MCBO` paper also contains potential evidence for this low-order hypothesis: a genetic algorithm, which only mutates a few variables at a time and does not inherently model high-order interactions, performs remarkably well on `RNA Inverse Folding`, significantly outperforming all BO methods. Lastly, we note that if such additive structure is known _a priori_, one can easily incorporate it into the heat-kernel, leading to methods such as (but not limited to) `RDUCB` and `CoCaBO` (see Appendix C.2).
>
> [1] Domingo, Júlia, Guillaume Diss, and Ben Lehner. _Pairwise and higher-order genetic interactions during the evolution of a tRNA._ Nature (2018).
>
> [2] Faure, Andre J., et al. _The genetic architecture of protein stability._ Nature (2024).
>
> >line 229: it should be "by choosing k", not "mathbb{k}"
>
> We acknowledge that our notation in the mentioned paragraph is confusing, and are thankful to the reviewer for bringing this to our attention. However, we do not believe that replacing "mathbb{$k$}" by "$k$" is correct. In the updated camera-ready version, we have now written: "by choosing mathbb{$k$}$(d)$ to be any (continuous) isotropic kernel (with $d = \lVert\mathbf{x} - \mathbf{x}'\rVert^2_2$), we always obtain a valid Hamming kernel $k$." We are happy to address any further questions the reviewer might have.
>
> >Have a line style distinct for heat-based and non-heat-based to ease the analysis
>
> We thank the reviewer for this great suggestion. We have made use of it for the camera-ready version of the paper, which now contains improved plots.
>
> >Is there a way to see the kernel from BODi as a heat kernel variation in some sense?
>
> We appreciate the reviewer's excellent question, and believe the answer is "no". In fact, we have constructed a short sketch for why `BODi` is not part of the Hamming-kernel class, and, as a consequence, cannot be seen as a (generalized) heat kernel. This proof has now been included in the appendix, and is referenced in the main part of the paper. We believe this is a meaningful addition to the paper, and thank the reviewer for highlighting this point.
>
> As outlined in Definition 6: a kernel is a Hamming kernel (`HK`) if and only if it depends on $\mathbf{x}, \mathbf{x}' \in \mathcal{X}$ through the square root of the Hamming distance $h(\mathbf{x}, \mathbf{x}')$. That  is,
> $$k_{\texttt{HK}}(\mathbf{x}, \mathbf{x}') := f\left(\sqrt{h(\mathbf{x}, \mathbf{x}')}\right),$$
> for some $f : \mathbb{Z}_0^+ \rightarrow \mathbb{R}$.
>
> In contrast, the `BODi` kernel depends on $\mathbf{x}, \mathbf{x}' \in \mathcal{X}$ through the term
> $$k_{\texttt{BODi}}(\mathbf{x}, \mathbf{x}') := g\left(\lVert\phi_{\mathbf{A}}(\mathbf{x})-\phi_{\mathbf{A}}(\mathbf{x}')\rVert^2_2\right),$$
> with
> $$g(d): =\left( 1 + \frac{\sqrt{5} d}{\ell} + \frac{5 d^2}{3 \ell^2} \right) \exp \left( -\frac{\sqrt{5} d}{\ell} \right).$$
> Here, $g(\cdot)$ is the Matérn-5/2 function, where we note that $g(\cdot)$ constitutes a bijective function. Additionally, $\phi_{\mathbf{A}}(\cdot)$ denotes the mapping towards the `HED` embedding (see section 4 of [3]), namely:
> $$\left[ \phi_\mathbf{A}(\mathbf{x}) \right]_i := h(\mathbf{a}_i, \mathbf{x}),$$
> with $\mathbf{a}_i \in \mathbf{A}$ one of the $M$ anchor points in dictionary $\mathbf{A}$.
>
> For `BODi` to be a Hamming kernel, there thus needs to exist a function, say $u(\cdot)$, such that
>  $$u\left(\sqrt{h(\mathbf{x}, \mathbf{x}')}\right) = \lVert\phi_{\mathbf{A}}(\mathbf{x})-\phi_{\mathbf{A}}(\mathbf{x}')\rVert^2_2.$$
> If this function $u(\cdot)$ exists, we can define $f:= g \circ u$, obtaining
> $$
> k_{\texttt{BODi}}(\mathbf{x}, \mathbf{x}') = g\left(\lVert\phi_{\mathbf{A}}(\mathbf{x})-\phi_{\mathbf{A}}(\mathbf{x}')\rVert^2_2\right)
> = g\left(u\left(\sqrt{h(\mathbf{x}, \mathbf{x}')}\right)\right)
> = f\left(\sqrt{h(\mathbf{x}, \mathbf{x}')}\right)
> = k_{\texttt{HK}}(\mathbf{x}, \mathbf{x}').
> $$
>
> However, as becomes clear by looking at the mapping $\phi_{\mathbf{A}}(\cdot)$, there does not exist a function $u(\cdot)$ satisfying the above claim. We provide a small counter-example with $M=1$ below.
>
> In scenario A, we have:
> $$\mathbf{x} = [1,1,1], \text{ } \mathbf{x'} = [1,1,2] \text{ and } \mathbf{a}\_1 = [1,1,1],$$
> which gives us
> $$\sqrt{h(\mathbf{x}, \mathbf{x}')} = 1 \text{ and } \lVert\phi_{\mathbf{A}}(\mathbf{x})-\phi_{\mathbf{A}}(\mathbf{x}')\rVert^2_2 = 1.$$
>
> In scenario B, we have:
> $$\mathbf{x} = [1,1,2], \text{ } \mathbf{x'} = [1,1,3] \text{ and } \mathbf{a}\_1 = [1,1,1],$$
> which gives us
> $$\sqrt{h(\mathbf{x}, \mathbf{x}')} = 1 \text{ and } \lVert\phi_{\mathbf{A}}(\mathbf{x})-\phi_{\mathbf{A}}(\mathbf{x}')\rVert^2_2 = 0.$$
>
> For $u(\cdot)$ to work in both scenarios, we would need to have:
> $$u(1) = 1 \text{ and } u(1) = 0,$$
> which is clearly not possible. As a result, we conclude that `BODi` is not a Hamming kernel, and therefore also not a heat kernel (or a direct generalization thereof).
>
> In fact, using the above proof-structure, we can show an even stronger result: `BODi` is not part of an (even broader) general class of kernels, which includes not only Hamming- and graph-based kernels, but also additive kernels such as `CoCaBO` and `RDUCB` (see Appendix C.2). With this, we feel confident in our answer that `BODi` cannot be seen as a heat-kernel variation in some sense.
>
> [3] Deshwal, Aryan, et al. _Bayesian Optimization over High-Dimensional Combinatorial Spaces via Dictionary-Based Embeddings._ AISTATS (2023).

---

> > ### Comment · Reviewer_opDa · 2025-08-04
> > **Thank you**
> >
> > Thank you! The paper is great. I am a big fan of the work. Perhaps the authors can elaborate on the findings related to my question in more detail in the main text and/or include the experiments.

---

> > > ### Author Response · Authors · 2025-08-09
> > > **Summary of updated manuscript + last two MCBO datasets**
> > >
> > > We sincerely thank the reviewer for their encouraging feedback and thoughtful responses. Our updated manuscript now includes all of the following elements:
> > > 1.  The two datasets (`RNA Inverse Folding` and `AIG Optimization`) have been added to the existing figures. Moreover, we have dedicated a new paragraph specifically to the performance of additive kernels on biological datasets, as well as our current hypotheses explaining this strong performance. We believe this is a relevant topic with real-world use cases, and appreciate the reviewer's efforts to meaningfully improving the paper.
> > > 2. Line 229 has been corrected with our proposed clarification (see previous response).
> > > 3. All plots have distinct line styles for heat-based and non-heat-based kernels.
> > > 4. The appendix contains a detailed proof, showing why `BODi` cannot be seen as different (generalized) classes of heat kernels. This proof is also referenced in the main paper.
> > >
> > > Additionally, since our last reply, we were able to run our experiments on the last two categorical datasets found in `MCBO`: `Antibody Design` and `MIG Optimization` (i.e. the two slow-simulator datasets). The new results match the ones described in our previous response: `MIG Optimization` is similar to `AIG Optimization`, and `Antibody Design` is similar to `RNA Inverse Folding` (with, again, strong performance for the additive kernels). These last two results have also been added to the final version of the paper, which now contains a total of 21 methods/kernels and 18 datasets (averaged for up to 80 seeds per method/kernel per dataset).

---

### Official Review · Reviewer_b9FW · 2025-07-03

**Clarity:** 3
**Significance:** 3
**Originality:** 4
**Rating:** 5
**Confidence:** 4

**Summary:**

This paper discusses algorithms for combinatorial Bayesian optimization (CBO), a problem that occurs in many practical applications. Specifically, this paper focuses on heat or diffusion kernels that operate on a graph that models the search space and Matern-class kernels (including the RBF kernel) relying on the Hamming distance.

The authors show that these Matern-class kernels _are_ heat-kernels if one considers categorical problems where the graph is modeled as a Hamming graph. Furthermore, they show that any Hamming kernel is a $\Phi$-kernel, which is a generalization of the Heat kernel.

Based on these insights, the authors propose a faster implementation of the heat kernel based on the Hamming distance.

The authors benchmark their kernel against various baselines on different categorical benchmark problems. They show that their simplified heat kernel is competitive with the state of the art. They also use random permutations of the categorical variables as proposed by Papenmeier et al. (2023) and show that the SSK kernel is also sensitive to those permutations (Papenmeier et al. have shown this for BODi and COMBO).

**Questions:**

* Why do we need Figures 2 and 4? Aren't they showing mostly the same thing?
* Do you have concrete ideas on how one could extend the analysis to ordinal problems?

**Ethical Concerns:**

["NO or VERY MINOR ethics concerns only"]

**Final Justification:**

As discussed under strengths, this paper addresses a relevant topic (combinatorial BO) and, for purely categorical problems, provides a computationally more efficient method of running BO with Heat kernels. This is possible due to an equivalence between using RBF kernels with Hamming distance and heat kernels discovered by the authors. Finally, the authors show that BOSS is sensitive to a permutation of categories, following up on a recent analysis of this issue.

Overall, this paper has several strong contributions but is focused on purely categorical problems - the analysis does not extend to, e.g., ordinal variables.

**Limitations:**

Yes

**Quality:**

3

**Strengths And Weaknesses:**

__Strengths__

* The paper is well written and follows a clear structure. The problem is relevant - categorical BO has various applications.
* Showing that an RBF kernel with Hamming distance is equivalent to COMBO reduces the computational complexity of running COMBO drastically.
* The authors show that yet another algorithm (BOSS) is sensitive to permutations of categories. This is highly relevant because categorical problems have no notion of 'order'. In most cases, it is arguably unexpected that randomly permuting the order of the categories affects performance, and the methods that are affected by this have not stated this implicit bias. Hence, it is arguably not by design.
* The authors are open about the fact that the analysis only holds for categorical problems.

__Weaknesses__

I see two main weaknesses. 1) The analysis only holds for categorical problems. 2) The proposed algorithm method is very close to already existing methods. I am not particularly concerned about 2) since the empirical performance is not the focus of this paper, and acts more as a sanity check. 1) is a bigger limitation, even though the authors argue that ordinal problems might not be as prevalent in practice.

---

> ### Author Rebuttal · Authors · 2025-07-31
>
> We are delighted to see the reviewer found the paper to be well-written, and the problem to be relevant. We appreciate  the reviewer's insights, and are happy to answer their questions.
>
> >Why do we need Figures 2 and 4? Aren't they showing mostly the same thing?
>
> Yes, that is correct: Figures 2 and 4 depict the same baselines, only on different datasets. The same is true for Figures 1 and 3. Due to space constraints, we decided to move some datasets to the appendix. In the camera-ready version, we have now more clearly described the relationship between these two pairs of figures (2/4 and 1/3). We are thankful to the reviewer for pointing out this possible point of confusion.
>
> >Do you have concrete ideas on how one could extend the analysis to ordinal problems?
>
> On line 139 of our submitted paper, we explain how to transform the discrete domain $\mathcal{X}$ into a graph $\mathcal{G}$. Specifically, we define the graphs $\mathcal{G}_i$ as complete graphs, and argue that this is a natural choice when the underlying variables $x_i \in \mathcal{X}_i$ are categorical variables. However, for other types of discrete variables, the graphs $\mathcal{G}_i$ can (and probably should) have different structures (as hinted at on line 143). For example, for an ordinal variable $x_i \in \mathcal{X}_i$, it is natural to define $\mathcal{G}_i$ as a path graph, where nodes are not equidistant anymore (whereas this is the case for complete graphs). In fact, this path-graph structure for ordinal variables is also suggested in the `COMBO` paper [1].
>
> Using the above idea, there is a clear way to connect ordinal variables to our unifying framework. First, define the graphs $\mathcal{G}_i$ according to the nature of their underlying variables $x_i \in \mathcal{X}_i$ (complete graph for categorical, path graph for ordinal, etc.). Second, create the graph $\mathcal{G}$ as the graph Cartesian product of all $\mathcal{G}_i$ (similar to what is done in Definition 2 of our paper). Lastly, apply the diffusion kernel from Equation 2 or 5 on this product-graph $\mathcal{G}$. This connection has now been made more explicit in the camera-ready-version, where we have greatly expanded our discussion regarding ordinal variables. We believe this is a meaningful addition, and are grateful to the reviewer for highlighting this aspect.
>
> Unfortunately, the above connection is also where the analysis stops. Since path graphs do not contain the same type of symmetries as complete graphs, the eigenvalues of the corresponding Laplacian do not simplify as much as for categorical variables. Therefore, we do not obtain a simple closed-form solution. In other words, Equation 2 or 5 cannot be simplified to something like Equation 3 or 4.
>
> Lastly, we would like to refer to our answer given to Reviewer `oZnX`, where we argue that most combinatorial BO papers do not include _truly_ ordinal variables. As a consequence, we cannot unify many well-known kernels into one unifying framework (as we did for categorical variables), since there simply do not exist many kernels designed for ordinal variables. We have now stressed this argument in the updated version of the paper.
>
> [1] Oh, Changyong, et al. _Combinatorial Bayesian Optimization using the Graph Cartesian Product._ NeurIPS (2019).

---

> > ### Comment · Reviewer_b9FW · 2025-08-02
> >
> > Thank you for the reply. I have already assigned a high score and will leave it as it is.

---

### Note · Authors · 2025-08-12

We would like to sincerely thank all reviewers for their commitment to bringing this paper to its best possible version. We received four high-quality reviews, replied to all of their comments and suggestions, and in return received a short reply from each reviewer stating that they were satisfied with our answer.

In this rebuttal period, a key contribution was the addition of more kernels and datasets, with the final version of the paper now having a total of 21 methods/kernels and 18 datasets (averaged for up to 80 seeds per method/kernel per dataset). Additionally, some theoretical holes were clarified (e.g. ordinal variables), and new proofs were added (e.g. `BODi` cannot be seen as a heat kernel).

Lastly, besides the reviewers' efforts to provide thoughtful feedback, we are also grateful for their encouraging words. As an example, Reviewer `opDa` kindly stated: "The paper is great. I am a big fan of the work".

---

### Decision · Program_Chairs · 2025-09-17

**Decision:**

Accept (poster)

**Comment:**

This paper considers the problem of Bayesian Optimization (BO) over combinatorial spaces, specifically the space of categorical variables. The key contribution of the paper is to show that Matern-class kernels including the RBF kernel are heat-kernels if one considers categorical problems where the graph is modeled as a Hamming graph (e.g., COMBO). The paper also shosw that any Hamming kernel is a $\Phi$-kernel, a generalization of the Heat kernel. The consequence of this understanding is that we have a computationally-efficient method of running BO with Heat kernels. The paper also show that BOSS (prior method) is sensitive to a permutation of categories building on a recent analysis. Experimental results are also strong.

All the reviewers' liked the paper and the author rebuttal/discussion further reinforced this positive opinion. I recommend accepting the paper and strongly encourage the authors' to incorporate all the discussion into the final paper and also release the code.

There are a couple of papers which explored some of the relevant concepts in this paper.
Equation 3 from the paper is the same as Equation 4.7 in HyBO paper (https://arxiv.org/abs/2106.04682)
Analytical form of COMBO was also explored in MerCBO paper (https://arxiv.org/pdf/2012.07762)